# G protein-specific mechanisms in the serotonin 5-HT$_{2A}$ receptor regulate psychosis-related effects and memory deficits

Elk Kossatz [1,14], Rebeca Diez-Alarcia [2,3,4,14], Supriya A. Gaitonde[5], Carla Ramon-Duaso [6], Tomasz Maciej Stepniewski[7,8], David Aranda-Garcia[7,9], Itziar Muneta-Arrate[2,3], Elodie Tepaz[5], Suwipa Saen-Oon [10], Robert Soliva[10], Aida Shahraki[11], David Moreira[12,13], Jose Brea[11,12], Maria Isabel Loza[12,13], Rafael de la Torre [1], Peter Kolb [11], Michel Bouvier [5], J. Javier Meana [2,3,4], Patricia Robledo [1] ✉ & Jana Selent [7,9] ✉

G protein-coupled receptors (GPCRs) are sophisticated signaling machines able to simultaneously elicit multiple intracellular signaling pathways upon activation. Complete (in)activation of all pathways can be counterproductive for specific therapeutic applications. This is the case for the serotonin 2 A receptor (5-HT$_{2A}$R), a prominent target for the treatment of schizophrenia. In this study, we elucidate the complex 5-HT$_{2A}$R coupling signature in response to different signaling probes, and its physiological consequences by combining computational modeling, in vitro and in vivo experiments with human postmortem brain studies. We show how chemical modification of the endogenous agonist serotonin dramatically impacts the G protein coupling profile of the 5-HT$_{2A}$R and the associated behavioral responses. Importantly, among these responses, we demonstrate that memory deficits are regulated by G$_{\alpha q}$ protein activation, whereas psychosis-related behavior is modulated through G$_{\alpha i1}$ stimulation. These findings emphasize the complexity of GPCR pharmacology and physiology and open the path to designing improved therapeutics for the treatment of stchizophrenia.

G protein-coupled receptors (GPCRs) are an important class of cell surface receptors. Due to their involvement in numerous physiological processes, they have become an important drug target class for numerous clinical indications, with 30-40% of marketed drugs acting through them[1]. However, the signaling complexity of GPCRs related to their ability to signal through numerous pathways, including disease-associated but also non-disease-associated pathways, have been linked to undesired side effects[2]. Moreover, the discovery that molecular modulators (so-called biased (ant)agonists) can bind to GPCRs and preferentially activate specific pathways over others (so-called pathway bias) has created the opportunity to improve drug development for complex diseases such as schizophrenia (SCZ)[3]. SCZ is a severe debilitating disease, characterized by positive symptoms (such as hallucinations), negative symptoms (including amotivation, anhedonia and alogia), and cognitive deficits[4]. Current antipsychotic medications, however, do not target cognitive or negative symptoms, which are the main factors contributing to the loss of functionality in SCZ patients[5]. In addition, most of the patients that

---

receive antipsychotic treatment suffer diverse side effects, and up to 30% of patients are resistant or respond only partially to treatment[6]. Therefore, there is a clear need for improving treatment strategies. Classical and atypical antipsychotic drugs have traditionally been designed to inhibit implicated GPCRs without considering the different intracellular signaling pathways triggered by the receptors. Such non-selective inhibition of all possible pathways associated with one GPCR target can result in reduced therapeutic efficacy and provoke unwanted side effects[7]. This seems to be the case for the serotonin (5-HT) 2 A receptor (5-HT$_{2A}$R)[8], a prominent target for the treatment of SCZ. Recent studies have shown that inhibition of 5-HT$_{2A}$R-mediated pathways by antipsychotic drugs results in an unwanted silencing of the metabotropic glutamate receptor 2 (mGlu$_2$R) transcription, suggesting that full receptor inactivation could be counterproductive for the treatment of SCZ[8]. Hence, selectively modulating only pathway(s) linked to the disease is a promising approach to obtaining more efficacious and safer drugs. However, the contribution of specific 5-HT$_{2A}$R-initiated pathways to SCZ-like symptoms (i.e. positive, negative and cognitive deficits) via the engagement of various subtypes of G$_\alpha$ protein subunits, i.e. G$_{\alpha q}$ or G$_{\alpha i}$ proteins, and β-arrestins[3,9–12] has remained largely elusive.

In this study, we successfully address the challenges of disentangling the role of 5-HT$_{2A}$R-mediated pathways in SCZ-like behavioral responses by applying a multidisciplinary approach. Using small molecular probes derived from the natural agonist 5-HT, we first reveal the complex 5-HT$_{2A}$R coupling signature across its different downstream transducers (G$_{\alpha q}$, G$_{\alpha i}$, β-arrestins 1 and 2) using live-cell bioluminescence resonance energy transfer (BRET)-based biosensors. Then, linking postmortem brain experiments with in vivo behavioral responses, we provide evidence that distinct 5-HT$_{2A}$R-mediated pathways are implicated in psychosis-related effects and memory deficits. These findings have been validated with pharmacological and genetic tools to determine G protein implication. Finally, molecular modeling and dynamics simulation of the binding of signaling probes with differential behavioral responses highlight the structural features underlying the different actions of 5-HT$_{2A}$R. Importantly, our findings have key implications for exploiting G protein-specific mechanisms not only for the design of a novel class of drugs with improved therapeutic profiles for the treatment of SCZ, but can also contribute to a better understanding of the disease etiology.

## Results

### BRET experiments reveal the complex spectrum of 5-HT$_{2A}$R signaling in living cells

To gain insight into the molecular determinants that drive ligand-mediated 5-HT$_{2A}$R signaling bias, we studied the engagement of the receptor's proximal effectors (G proteins and β-arrestins) upon stimulation with structurally closely related probes of the endogenous agonist 5-HT (Fig. 1), including Met-I (3-(2-aminoethyl)−1-methyl-1H-indol-5-ol hydrochloride), Nitro-I (2-(5-nitro-1H-indol-3-yl)ethamine hydrochloride), OTV1 (2-[5-(2,3-dihydro-1,4-benzodioxin-6-yl)−1H-indol-3-yl]ethan-1-amine) and OTV2 (2-(5-phenoxy-1H-indol-3-yl)ethan-1-amine)). The two latter compounds have been obtained from a virtual screen (see method section). Competition binding experiments

with [³H]-ketanserin confirm that the test compounds bind to the orthosteric binding site of the 5-HT$_{2A}$R (Fig. 1).

In a first screen, we explored the coupling spectrum of the 5-HT$_{2A}$R to several G$_\alpha$ protein subtypes and β-arrestins upon stimulation with Nitro-I, Met-I, OTV1 and OTV2 in living HEK-293 cells using BRET-based assays (Fig. 2A). The pEC$_{50}$, Emax and log τ/$K_A$ for G$_{\alpha q}$, G$_{\alpha 11}$, G$_{\alpha 14}$, G$_{\alpha 15}$ G$_{\alpha i1}$, G$_{\alpha i2}$, G$_{\alpha i3}$, G$_{\alpha oA}$, G$_{\alpha oB}$, and G$_{\alpha z}$ are listed in Table 1, whereas full concentration response curves are shown only for representative members of the G$_{\alpha q}$ and G$_{\alpha i}$ family and β-arrestin 1 and 2 in Fig. 2B–G. We find that all compounds are full agonists toward the canonical G$_{\alpha q}$ pathway, but only partial agonists on the G$_{\alpha i}$ family and βarrs when compared to 5-HT. Notably, among the G$_{\alpha i}$ family members, the relative efficacy is greater toward G$_{\alpha i1}$ vs G$_{\alpha i2}$, or G$_{\alpha i3}$. To further characterize the signaling profiles promoted by the 5-HT analogs, the transduction coefficient (log τ/$K_A$) was determined for each pathway using the operational model[13,14] followed by calculation of the ligand-physiology bias[7] factor between pathways using the endogenous agonist 5-HT as the reference.

Figure 2H illustrates the coupling preference and physiology-bias profile promoted by each of the compounds as compared to 5-HT. Nitro-I and Met-I show a general physiology bias for the canonical G$_{\alpha q}$ over G$_{\alpha i1}$, G$_{\alpha i2}$, G$_{\alpha i3}$, β-arrestin 1, and 2 (light to dark blue, Fig. 2H). Of note is the high magnitude of G$_{\alpha q}$ bias over β-arrestin 2 for both compounds with 10 (Nitro-I) and > 50 fold (Met-I), and the high G$_{\alpha q}$ bias over G$_{\alpha i1}$ with > 50 (Nitro-I) and 8-fold bias (Met-I).

Interestingly, introducing an extended substituent (i.e. phenoxy substituent) in position 5 of the indole fragment (Fig. 1), as in OTV2, results in a distinct coupling (Fig. 2B–G) and bias profile of 5-HT$_{2A}$R (Fig. 2H) compared to Nitro-I and Met-I. OTV2 induces a general bias toward the G$_{\alpha i}$ family (G$_{\alpha i1}$, G$_{\alpha i2}$, G$_{\alpha i3}$) over G$_{\alpha q}$ (light to dark red colors, Fig. 2H) in comparison to 5-HT, which is mainly driven by the increased potency of OTV2 towards the G$_{\alpha i}$ family members. Furthermore, we observe that OTV2 has a substantially reduced ability to promote the recruitment of β-arrestin 1, yielding a 17-fold G$_{\alpha q}$ bias over β-arrestin 1 while the G$_{\alpha q}$ bias over β-arrestin 2 observed for Nitro-1 and Met-I was lost for OTV2 (Fig. 2H and Table 1).

Further extension of position 5 (i.e. 1,4-benzodioxin) (Fig. 1) produces the most dramatic changes in the coupling (Fig. 2B–G) and physiology-bias profile (Fig. 2H). OTV1 loses to a large extent its stimulating activity for G$_{\alpha i2}$, G$_{\alpha i3}$ as well as its ability to recruit β-arrestin 1 and 2 (Fig. 2D–G). This results in a very significant G$_{\alpha q}$ activation preference over G$_{\alpha i1}$, with a bias factor > 20 compared to 5-HT. An even greater preference is observed toward G$_{\alpha q}$ vs G$_{\alpha i2}$, G$_{\alpha i3}$, β-arrestin 1 and 2. However, virtually no activation of these pathways prevented us from calculating a formal bias factor (gray colored, Fig. 2H). Whether this preference results from a structurally-driven signaling bias or from the lower potency of OTV1 toward all pathways (i.e. activation of pathways that are not strongly coupled to the receptor are difficult to be detected) cannot be determined.

An interesting observation is that some of the observed pEC50 values are many orders of magnitude larger than the corresponding binding affinities (Fig. 1B). An example is Met-I with a pEC50 of 9.6 for G$_{\alpha q}$ activation (Table 1) versus a pKi of 5.5 for its binding affinity to the 5-HT$_{2A}$R (Fig. 1). This difference most likely results from the diverse

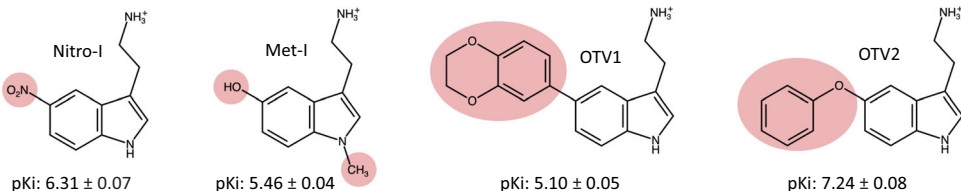

**Fig. 1 | Structural derivatives of the endogenous 5-HT$_{2A}$R agonist serotonin (5-HT).** Ligand binding affinities (pKi) are indicated for Nitro-I, Met-I, OTV1 and OTV2 obtained in [³H]ketanserin competition binding experiments in CHO cells (*n* = 3). The data represent the mean ± SD (see methods section).

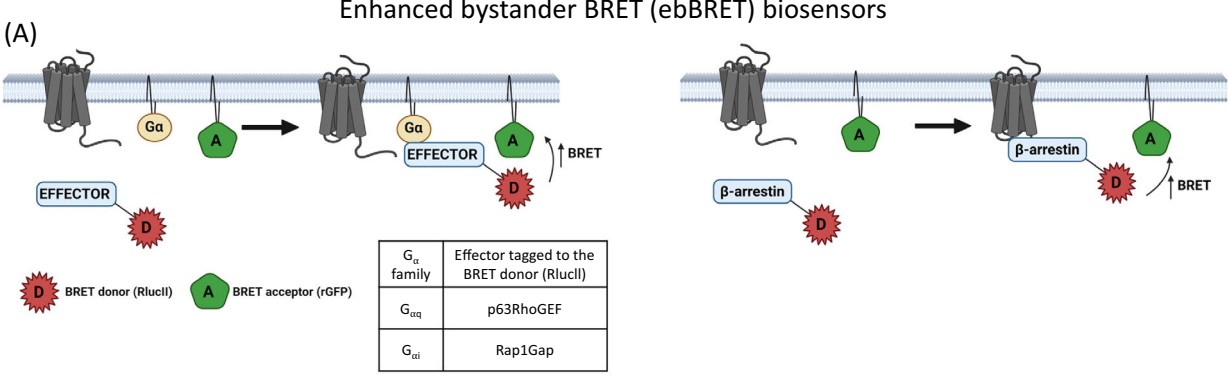

**Enhanced bystander BRET (ebBRET) biosensors**

**Concentration response curves for G protein pathways and recruitment of the β-arrestins**

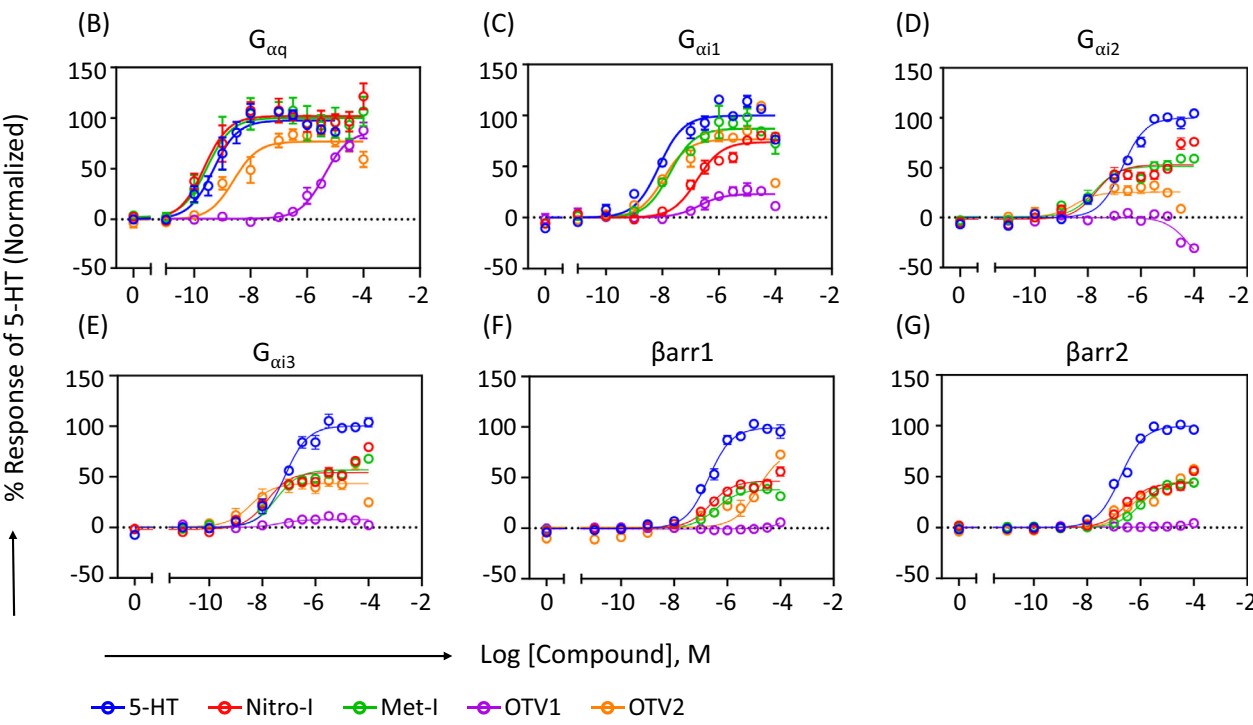

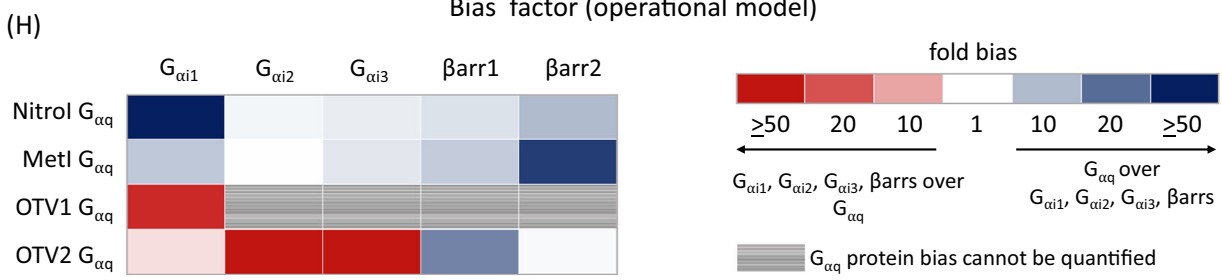

**Bias factor (operational model)**

**Fig. 2 | G protein coupling and β-arrestin recruitment for Nitro-I, Met-I, OTV1 and OTV2 in living cells. A** Schematic representation of enhanced bystander BRET (ebBRET) biosensors used to measure G protein activation and β-arrestin (βarr) recruitment in HEK-293 cells, created with BioRender. **B**–**G** Dose response curves depicting the activity of Nitro-I, Met-I, OTV1 and OTV2 at the different G protein signaling pathways and for the recruitment of βarr 1 and 2. The ligand-promoted BRET (ΔBRET) was further normalized with respect to the response of 5-HT (mean ± SEM; $n = 3$). **H** Bias factor for the $G_{\alpha q}$, $G_{\alpha i}$ family and βarr1 and 2. The operational model was used for bias calculations (see method section in supplementary information).

**Table 1 | Functional selectivity profile of Nitro-I, Met-I, OTV1, and OTV2 from BRET assays in the agonist mode**

| | 5-HT | | | Nitro-I | | | Met-I | | | OTV1 | | | OTV2 | | |
|---|---|---|---|---|---|---|---|---|---|---|---|---|---|---|---|
| | $pEC_{50}$ | $E_{max}$ | $log\tau/K_A$ | $pEC_{50}$ | $E_{max}$ | $log\tau/K_A$ | $pEC_{50}$ | $E_{max}$ | $log\tau/K_A$ | $pEC_{50}$ | $E_{max}$ | $log\tau/K_A$ | $pEC_{50}$ | $E_{max}$ | $log\tau/K_A$ |
| $G\alpha_q$ | 9.75 ± 0.23 | 101.10 ± 2.37 | 9.70 ± 0.15 | 9.65 ± 0.19 | 102.50 ± 9.81 | 9.67 ± 0.20 | 9.55 ± 0.31 | 98.16 ± 12.19 | 9.57 ± 0.39 | 5.38 ± 0.08 | 86.62 ± 3.72 | 5.28 ± 0.11 | 8.35 ± 0.19 | 75.20 ± 5.12 | 8.24 ± 0.14 |
| $G\alpha_{11}$ | 10.26 ± 0.08 | 98.03 ± 3.46 | 9.46 ± 0.06 | 9.96 ± 0.23 | 78.67 ± 6.99 | 9.13 ± 0.13 | 8.88 ± 0.11 | 79.73 ± 3.08 | 8.60 ± 0.12 | 5.41 ± 0.10 | 127.30 ± 11.26 | 5.93 ± 0.15 | 9.56 ± 0.12 | 80.31 ± 3.35 | 8.92 ± 0.12 |
| $G\alpha_{14}$ | 9.30 ± 0.10 | 98.44 ± 5.21 | 9.14 ± 0.09 | 8.72 ± 0.15 | 102.90 ± 5.67 | 8.73 ± 0.14 | 8.72 ± 0.11 | 98.77 ± 4.13 | 8.72 ± 0.13 | 5.72 ± 0.13 | 94.48 ± 5.94 | 5.64 ± 0.15 | 9.18 ± 0.25 | 90.79 ± 6.32 | 9.13 ± 0.14 |
| $G\alpha_{15}$ | 8.10 ± 0.08 | 99.36 ± 3.22 | 8.03 ± 0.09 | 7.61 ± 0.13 | 107.80 ± 4.62 | 7.66 ± 1.03 | 7.22 ± 0.27 | 98.48 ± 5.40 | 7.21 ± 0.10 | NA | NA | NA | 7.55 ± 0.33 | 60.05 ± 6.63 | 7.68 ± 0.04 |
| $G\alpha_{i1}$ | 8.12 ± 0.17 | 99.92 ± 6.02 | 8.16 ± 0.15 | 6.78 ± 0.09 | 73.92 ± 2.88 | 6.42 ± 0.15 | 7.73 ± 0.15 | 87.12 ± 4.55 | 7.56 ± 0.15 | 6.71 ± 0.28 | 23.21 ± 2.96 | 5.59 ± 0.44 | 7.96 ± 0.25 | 76.81 ± 6.93 | 7.65 ± 0.18 |
| $G\alpha_{i2}$ | 6.71 ± 0.08 | 99.95 ± 3.55 | 6.70 ± 0.15 | 7.69 ± 0.22 | 54.55 ± 4.28 | 6.73 ± 0.61 | 7.75 ± 0.16 | 50.40 ± 2.84 | 6.90 ± 0.60 | NA | | | 8.50 ± 0.38 | 25.05 ± 3.43 | 7.23 ± 0.70 |
| $G\alpha_{i3}$ | 7.16 ± 0.10 | 100 ± 3.89 | 7.20 ± 0.10 | 7.91 ± 0.15 | 54.48 ± 3.07 | 7.07 ± 0.18 | 7.47 ± 0.13 | 56.76 ± 2.50 | 6.88 ± 0.17 | NA | | | 8.45 ± 0.31 | 42.55 ± 4.58 | 7.70 ± 0.23 |
| $G\alpha_{oA}$ | 7.02 ± 0.08 | 99.99 ± 3.33 | 7.01 ± 0.15 | 7.27 ± 0.24 | 51.01 ± 4.45 | 6.64 ± 0.51 | 7.14 ± 0.19 | 47.38 ± 3.50 | 6.25 ± 0.33 | NA | | | 8.35 ± 0.25 | 29.67 ± 2.68 | 7.76 ± 0.51 |
| $G\alpha_{oB}$ | 6.98 ± 0.08 | 99.85 ± 3.46 | 7.05 ± 0.11 | 7.25 ± 0.23 | 45.76 ± 3.87 | 6.57 ± 0.24 | 7.12 ± 0.16 | 47.46 ± 2.89 | 6.63 ± 0.23 | NA | | | 8.04 ± 0.45 | 40.55 ± 6.64 | 7.18 ± 0.27 |
| $G\alpha_z$ | 8.00 ± 0.08 | 94.48 ± 3.32 | 8.13 ± 0.08 | 7.36 ± 0.14 | 100.30 ± 5.06 | 8.69 ± 0.13 | 7.20 ± 0.13 | 82.95 ± 4.19 | 8.55 ± 0.10 | 8.12 ± 1.81 | 16.44 ± 1.52 | 4.67 ± 0.90 | 8.12 ± 0.21 | 61.83 ± 5.17 | 7.89 ± 0.31 |
| β-arrestin1 | 6.71 ± 0.06 | 99.31 ± 2.59 | 6.88 ± 0.05 | 6.65 ± 0.09 | 46.59 ± 1.95 | 6.32 ± 0.14 | 6.44 ± 0.08 | 38.18 ± 1.40 | 6.04 ± 0.17 | NA | | | 4.80 ± 0.17 | 76.65 ± 9.22 | 4.37 ± 0.10 |
| β-arrestin2 | 6.75 ± 0.04 | 99.70 ± 1.85 | 6.84 ± 0.05 | 6.55 ± 0.11 | 43.78 ± 2.15 | 6.15 ± 0.12 | 6.02 ± 0.05 | 44.96 ± 1.10 | 5.63 ± 0.12 | NA | | | 6.32 ± 0.15 | 45.50 ± 3.22 | 5.44 ± 0.13 |

*NA* no activity. Data represent mean ± SEM of 3 independent experiments.

experimental setups including the expression level of receptors or G proteins[15]. Interestingly, additional BRET experiments show that receptor expression levels do not affect the observed pEC50 values (Fig. S9). In contrast, we find that increasing levels of $G_{\alpha q}$ significantly increment the apparent potency of Met-I (Fig. S10). This finding demonstrates that the expression level of G protein is a critical parameter that largely contributes to the differences observed between pEC50 of G protein activation and pKi of ligand binding. It further underscores the difficulties associated with comparing data points across distinct experimental setups and could also explain the differences observed between cell-based and ex vivo experiments.

## Met-I and OTV1 reduce the basal activity of the $G_{\alpha i1}$ pathway via the 5-HT$_{2A}$R in postmortem brain tissue

Next, we explored the ability of our molecular probes to modulate the activity of diverse G protein subtypes ($G_{\alpha q/11}$, $G_{\alpha i1}$, $G_{\alpha i2}$, $G_{\alpha i3}$) in postmortem human brain tissue. For this, we used [$^{35}$S]GTPγS binding experiments coupled to immunoprecipitation of $G_\alpha$ subunits of heterotrimeric G proteins (Fig. 3A) in postmortem human dorsolateral prefrontal cortex (PFC) membrane-enriched fractions[16,17]. To confirm that these effects were mediated through 5-HT$_{2A}$R, the same assays were carried out in the presence of MDL-11,939, a 5-HT$_{2A}$R-selective neutral antagonist[18] (Fig. 3B–E, Fig. S1, Table 2).

Our study reveals that Nitro-I triggered statistically significant activation of $G_{\alpha i1}$, $G_{\alpha i3}$ and $G_{\alpha q/11}$. Among them, only $G_{\alpha i1}$ and $G_{\alpha q/11}$ are 5-HT$_{2A}$R-mediated, as co-incubation with MDL-11,939 reversed the observed effect completely or partially, respectively (Fig. 3B, Table 2). Met-I modulated the activity of all studied $G_\alpha$ subunit subtypes, with the exception of $G_{\alpha i2}$. Selective 5-HT$_{2A}$R inhibition with MDL-11,939 suggests that inverse agonism at $G_{\alpha i1}$ and agonism at $G_{\alpha i3}$, $G_{\alpha q/11}$ are directly mediated by 5-HT$_{2A}$R (Fig. 3C, Table 2). In the same way, although OTV1 and OTV2 are able to modulate all studied $G_\alpha$ subunit subtypes (Fig. 3D, E, Table 2), only $G_{\alpha i1}$, $G_{\alpha i3}$ and $G_{\alpha q/11}$ modulation is 5-HT$_{2A}$R-mediated, but not $G_{\alpha i2}$ modulation. However, one main difference is related to the observation that Met-I and OTV1 elicit inverse agonism at the $G_{\alpha i1}$ whereas Nitro-I and OTV2 show a $G_{\alpha i1}$ agonism. The overall observed $G_{\alpha q/11}$ activation for tested compounds in postmortem brain samples (Fig. 3B–E) is in line with the activation of the canonical $G_{\alpha q}$ in our BRET experiments in living cells (Fig. 2D). Interestingly, differences are found for the regulation of $G_{\alpha i1}$, $G_{\alpha i2}$ and $G_{\alpha i3}$ subunit's activity between postmortem brain samples and the cell-based setup. These differences are not surprising considering variations in the experimental environment between postmortem brain samples and cell-based assays. This includes different expression levels of G proteins, 5-HT$_{2A}$R and other GPCRs. In fact, the presence of other GPCR types has been reported to promote the formation of heteromers which can alter the coupling response of 5-HT$_{2A}$R. For instance, signaling via the CB$_1$R-5-HT$_{2A}$R heteromer promotes $G_{\alpha i}$ coupling and not the canonical $G_{\alpha q}$ coupling of the 5-HT$_{2A}$R[19]. The presence of other factors in the postmortem tissue *vs* the cell line system could also explain the difference. The fact that Met-I and OTV1 are inverse agonists on $G_{\alpha i1}$ in the human brain but partial agonists for this pathway in the cell-based assays could be related to a higher basal tone for the 5-HT$_{2A}$R-promoted $G_{\alpha i1}$ activation in the tissue.

To further demonstrate the role of 5-HT$_{2A}$R in the observed effects in postmortem human brains, tissue homogenates from the brain cortex of wild type (WT) and 5-HT$_{2A}$R knockout (KO) mice were incubated with Nitro-I, Met-I, OTV1, and OTV2 in [$^{35}$S]GTPγS binding experiments (Fig. 3F–I, Fig. S2, Table S1). Importantly, these experiments confirm 5-HT$_{2A}$R-mediated signaling profiles observed in postmortem human PFC. Only small differences are found for the Nitro-I-induced $G_{\alpha q/11}$ activation, which is exclusively mediated through the 5-HT$_{2A}$R in mice, whereas a partial blockade was observed in human brain with the 5-HT$_{2A}$R-selective antagonist MDL-11,939.

All in all, our experiments highlight the complex signaling profile elicited by the tested compounds (Fig. 3B–I) that behave as agonists, inverse agonists or show no effect for the different subunit subtypes in homogenates from human PFC or mice brain cortex. Furthermore, we find that the observed coupling profile is often the result of interaction with multiple receptor types, as selective 5-HT$_{2A}$R antagonism does not always reverse the observed effects. This is also indicated by the finding that some of the effects are still present in 5-HT$_{2A}$R KO mice brain tissue. Most importantly, we identify two compounds, Met-I and OTV1, that show an inverse agonism effect over the $G_{\alpha i1}$ via the 5-HT$_{2A}$R. Since different guanosine diphosphate (GDP) concentrations and specific antibodies are used for the detection of each $G_\alpha$ subunit, results obtained from this methodological approach are semiquantitative. Thus, no quantitative comparisons can be made between the different studied subunits, no bias factor can be calculated, and only the effects of different compounds over the same subunit can be compared.

## $G_{\alpha i}$ agonism is implicated in psychosis-related effects

To interrogate the implication of specific 5-HT$_{2A}$R-mediated pathways in psychosis-related behavior, we investigated the effects of in vivo administration of our test compounds in mice on the head twitch response (HTR). The HTR serves as a behavioral proxy in rodents for human psychedelic effects, and can be used to discriminate hallucinogenic and non-hallucinogenic 5-HT$_{2A}$R agonists[9,20–25]. Thus, increasing doses of Nitro-I, Met-I, OTV1 or OTV2 were administered intracerebroventricularly (ICV), as well as DOI, a classic psychedelic 5-HT$_{2A}$R agonist, chosen as control. In a first step, we confirmed that our test compounds are not lethal or produced any physiological or neurotoxicity symptoms in mice for tested doses by the Irwin test (see Methods).

As expected, the HTR is significantly increased by our reference compound DOI with respect to vehicle-treated animals (Fig. S3, inset). We found that Nitro-I and OTV2, compounds triggering a

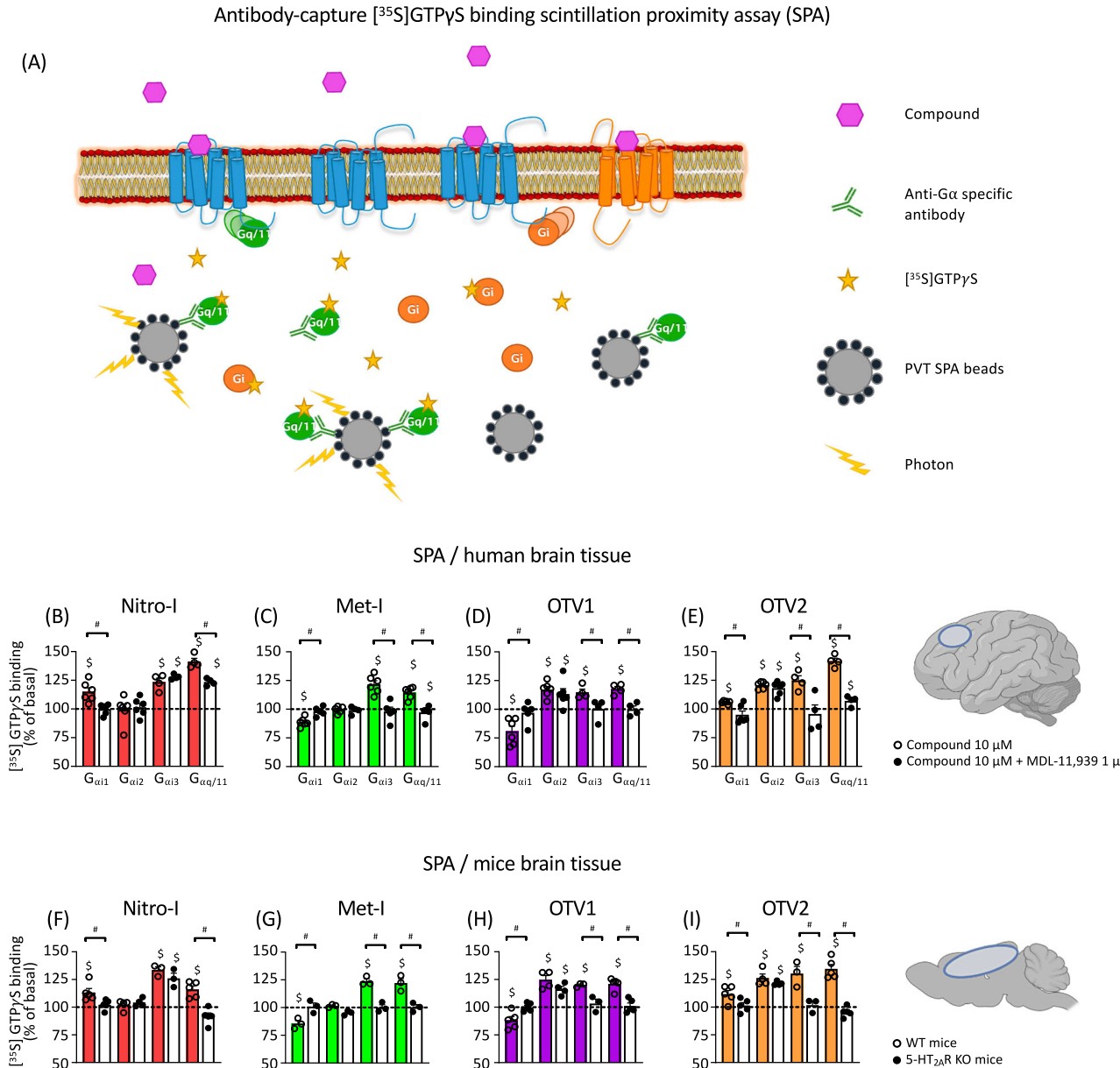

**Fig. 3 | Antibody-capture [35S]GTPγS binding scintillation proximity assay (SPA). A** Schematic representation of the SPA methodology, created using BioRender. SPA allows the determination of the activation level of different $G_\alpha$ subunit subtypes present in postmortem brain tissue thanks to the selective immunoprecipitation of each of them and their coupling to protein A-coated polyvinyltoluene SPA beads. Modulation of specific [35S]GTPγS binding to $G_{\alpha i1}$, $G_{\alpha i2}$, $G_{\alpha i3}$ and $G_{\alpha q/11}$ proteins by 10 μM Nitro-I (**B**), Met-I (**C**), Otava 3575001 (OTV1) (**D**) and Otava 3736689 (OTV2) (**E**) in human prefrontal cortex both in the absence (colored bars) and in the presence (white bars) of the 5-HT$_{2A}$R antagonist MDL-11,939. Basal values of specific [35S]GTPγS binding to the different G proteins are expressed as 100% and stimulatory/inhibitory effects on the respective basal are shown. Individual dots represent different assays 4−6 carried out for each subunit/condition performed in duplicate or triplicate. White dots represent data for the drug-alone condition, and black dots represent data for the drug + MDL condition (**B**−**E**). #$p < 0.05$ vs 100%; $$p < 0.05$ vs incubation in the presence of 1 μM MDL-11,939 ($t$-test). Modulation of specific [35S]GTPγS binding to $G_{\alpha i1}$-, $G_{\alpha i2}$-, $G_{\alpha i3}$- and $G_{\alpha q/11}$ proteins by 10 μM Nitro-I (**F**), Met-I (**G**), OTV1 (**H**) and OTV2 (**I**) in brain cortex tissue from WT (colored bars) and 5-HT$_{2A}$R KO mice (white bars). #$p < 0.05$ vs 100%; $$p < 0.05$ WT vs 5-HT$_{2A}$R KO ($t$-test). Basal values of specific [35S]GTPγS binding to the different $G_\alpha$ proteins are expressed as 100%, and stimulatory/inhibitory effects are expressed as % of the respective basal activity. Individual dots represent the different assays (3−5) for each subunit performed in duplicate or triplicate. White dots represent data for WT mice and black dots represent data for KO mice (**F**−**I**).

5-HT$_{2A}$R-mediated $G_{\alpha i1}$, $G_{\alpha i3}$ and $G_{\alpha q/11}$ agonism in postmortem brain assays, significantly increased HTR as compared to vehicle administration (Fig. S3A, D). Importantly, this effect is absent in 5-HT$_{2A}$R KO mice (Fig. 4A, D), indicating a 5-HT$_{2A}$R-dependent mechanism, and supporting other data showing that the 5-HT$_{2A}$R is necessary for the expression of HTR[21].

Interestingly, Met-I and OTV1, which showed 5-HT$_{2A}$R-mediated $G_{\alpha i1}$ inverse agonism in brain tissue experiments, do not increase HTR at any

of the doses tested (Fig. S3B, C) in WT or 5-HT$_{2A}$R KO mice (Fig. 4B, C). Note that in addition to inverse agonism of $G_{\alpha i1}$, both compounds also stimulate the activity of $G_{\alpha i3}$ and $G_{\alpha q}$ subtypes. To evaluate the possible involvement of $G_{\alpha i1}$, $G_{\alpha i3}$ and $G_{\alpha q}$ in the regulation of psychosis-related effects (i.e. HTR) through 5-HT$_{2A}$R stimulation, we used two different methodological approaches. On one hand, we carried out pharmacological inhibition of $G_{\alpha q/11}$ by ICV administration of YM-254890. On the other hand, we reduced the expression level of $G_{\alpha i1}$ and $G_{\alpha i3}$ genes,

**Table 2 | Normalized [$^{35}$S]GTPγS binding values for $G_{\alpha i1}$, $G_{\alpha i2}$, $G_{\alpha i3}$, and $G_{\alpha q/11}$ in postmortem human prefrontal cortex membrane homogenates**

| | Nitro-I | | | | | Nitro-I + MDL | | | | | p value | Met-I | | | | | Met-I + MDL | | | | | p value |
|---|---|---|---|---|---|---|---|---|---|---|---|---|---|---|---|---|---|---|---|---|---|---|
| | Mean | ± | SEM | n | p value | Mean | ± | SEM | n | p value | | Mean | ± | SEM | n | p value | Mean | ± | SEM | n | p value | |
| Gαi1 | 114.9 | ± | 3.5 | 6 | **0.008** | 100.0 | ± | 1.8 | 6 | ns | *0.006* | 88.7 | ± | 1.6 | 6 | **0.001** | 98.2 | ± | 1.8 | 5 | ns | *0.004* |
| Gαi2 | 99.2 | ± | 4.8 | 6 | ns | 101.5 | ± | 3.2 | 6 | ns | *ns* | 99.5 | ± | 1.4 | 5 | ns | 99.0 | ± | 1.6 | 4 | ns | *ns* |
| Gαi3 | 123.4 | ± | 3.4 | 4 | **0.006** | 127.9 | ± | 1.0 | 4 | **<0.001** | *ns* | 121.6 | ± | 3.2 | 6 | **0.001** | 98.0 | ± | 3.8 | 5 | ns | *0.001* |
| Gαq/11 | 141.3 | ± | 2.9 | 4 | **0.001** | 123.5 | ± | 1.5 | 4 | **0.001** | *0.004* | 114.4 | ± | 2.1 | 6 | **0.001** | 97.2 | ± | 2.7 | 5 | ns | *0.001* |

| | OTV1 | | | | | OTV1 + MDL | | | | | p value | OTV2 | | | | | OTV2 + MDL | | | | | p value |
|---|---|---|---|---|---|---|---|---|---|---|---|---|---|---|---|---|---|---|---|---|---|---|
| | Mean | ± | SEM | n | p value | Mean | ± | SEM | n | p value | | Mean | ± | SEM | n | p value | Mean | ± | SEM | n | p value | |
| Gαi1 | 80.6 | ± | 4.6 | 6 | **0.008** | 97.0 | ± | 4.2 | 5 | ns | *0.027* | 105.9 | ± | 0.5 | 6 | **<0.0001** | 95.0 | ± | 3.2 | 6 | ns | *0.018* |
| Gαi2 | 117.2 | ± | 2.7 | 6 | **0.001** | 113.3 | ± | 4.7 | 6 | **0.036** | *ns* | 120.8 | ± | 1.2 | 6 | **<0.0001** | 118.2 | ± | 2.3 | 6 | **0.001** | *ns* |
| Gαi3 | 114.7 | ± | 2.8 | 4 | **0.013** | 100.3 | ± | 4.6 | 4 | ns | *0.044* | 124.9 | ± | 4.1 | 4 | **0.009** | 95.7 | ± | 7.8 | 4 | ns | *0.024* |
| Gαq/11 | 117.6 | ± | 2.3 | 4 | **0.005** | 100.0 | ± | 2.8 | 4 | ns | *0.003* | 141.7 | ± | 2.6 | 4 | **0.001** | 107.2 | ± | 2.2 | 4 | **0.045** | *<0.0001* |

An agonist behavior results in a significant increase over basal binding, while an inverse agonist reduces it and an antagonist would not modify it. Results were analyzed by two-tailed Student's t-test (one-sample) vs basal values (expressed as 100%) or by two-tailed Student's t-test (two-sample) between conditions (presence vs absence of MDL-11,939; italicized p values). p values under 0.05 are highlighted in bold. Data are described as mean ± SEM values. ns: non-significant.

GNAI1 and GNAI3 respectively, by the ICV administration of specific antisense oligonucleotides (ODNs). OTV2 was selected as a model drug for these experiments, as it induces HTR and shows a 5-HT$_{2A}$R-mediated activation of $G_{\alpha i1}$, $G_{\alpha i3}$ and $G_{\alpha q}$ in postmortem brain tissue.

Importantly, we find that OTV2-induced HTR at the dose of 0.05 μg/μl was not modulated by $G_{\alpha q/11}$ inhibition using YM-254890 (Fig. 4E). Instead, decreasing protein levels for both $G_{\alpha i1}$ and $G_{\alpha i3}$ using ODNs abrogated the OTV2-induced HTR (Fig. 4F). We confirmed that $G_{\alpha i1}$ and $G_{\alpha i3}$ protein levels were significantly decreased in mice after chronic treatment with $G_{\alpha i1}$- or $G_{\alpha i3}$-ODNs compared to ODN-RDN (i.e., a random oligo) using Western blot analysis (Fig. S4), whereas no change was observed for the expression levels of $G_{\alpha q/11}$. The specificity of the used antibodies has been previously demonstrated[16]. Surprisingly, although the used ODNs had been previously described in the literature[26], a cross-effect of $G_{\alpha i1}$ ODN and $G_{\alpha i3}$ ODN treatment over both $G_{\alpha i1}$ and $G_{\alpha i3}$ protein expression levels was observed in our hands (Figure S4). Therefore, we are not able to discriminate between $G_{\alpha i1}$- and $G_{\alpha i3}$- mediated effects with the currently existing reagents.

Altogether, our results provide evidence that the activation of $G_{\alpha i}$ protein family-coupled signaling pathways ($G_{\alpha i1}$ and/or $G_{\alpha i3}$) via the 5-HT$_{2A}$R is a main contributor to psychosis-related effects in mice. Although our data suggest that $G_{\alpha i1}$-activation is necessary for this effect, we cannot completely exclude mechanisms other than $G_{\alpha i/o}$ activation in mediating HTR. Previous studies describe the involvement of other coupling partners including $G_{\alpha q}$[25,27–30], although there are also studies showing that $G_{\alpha q}$ KO mice inhibit only partially HTR[27], suggesting additional contributing mechanisms in HTR. Moreover, studies have reported the involvement of $G_{\alpha s}$ proteins[30], $G_{\beta \gamma}$ subunits[31], and β-arrestins[11,12] in HTR.

**Long-term memory performance is linked to 5-HT$_{2A}$R-induced $G_{\alpha q}$ activation**

To investigate the ability of our compounds to modulate cognitive performance via the 5-HT$_{2A}$R, we carried out the novel object recognition (NOR) test in WT and 5-HT$_{2A}$R KO mice. Indeed, we find that Met-I and OTV1 induce significant long-term memory deficits in WT mice at both doses tested through a 5-HT$_{2A}$R-dependent mechanism (Fig. 4H, I). Interestingly, OTV2 produces long-term memory deficits in WT mice only at the lower dose, and this effect is absent in KO mice (Fig. 4J). To our surprise, we find that Nitro-I is the only compound that does not induce long-term memory deficits (Fig. 4G).

To interrogate which pathway is associated with these effects over cognition, we chose again OTV2, our model compound able to induce long-term cognitive impairment in addition to HTR. For discriminating which of the $G_{\alpha}$ protein subtypes ($G_{\alpha i1}$, $G_{\alpha i3}$ and $G_{\alpha q}$) activated by OTV2

through a 5-HT$_{2A}$R-mediated mechanism in human and mouse brain tissue are implicated in these memory effects, we used again the two approaches previously described in the HTR section: (i) reduction of protein expression levels of $G_{\alpha i1}$ and $G_{\alpha i3}$ via ODN administration and (ii) $G_{\alpha q}$ pharmacological inhibition using YM-254890.

We find that reducing the protein levels of $G_{\alpha i1}$ and $G_{\alpha i3}$ via ODNs did not influence long-term memory deficits induced by OTV2 at the dose of 0.025 μg/μl (Fig. 4L). Instead, inhibiting $G_{\alpha q}$ activation with YM-254890 abrogated OTV2-induced long-term memory deficits (Fig. 4K). These results provide first evidence that long-term memory performance is linked to modulation of the $G_{\alpha q}$-coupled pathway via the 5-HT$_{2A}$R. Importantly, our data suggest further that cognitive deficits require other co-factors/events in addition to $G_{\alpha q}$ activation. This can be concluded from the observation that although Nitro-I also activates $G_{\alpha q}$ in cell-based and brain tissue experiments, it does not elicit cognitive deficits.

In a final experiment, we also evaluated whether our compounds induced anhedonia, one of the features of depression, and our results showed that none of the doses administered acutely evoked this behavior in WT or KO mice (Fig. S5).

**Ligands with differential 5-HT$_{2A}$R coupling profiles and in vivo responses establish distinct ELC2 interactions**

In the previous experiments, we have employed signaling probes that are structurally closely related to the endogenous neurotransmitter 5-HT. One main difference between these signaling probes are diverse substituents in position 5 of the indole scaffold, which proved to alter the 5-HT$_{2A}$R coupling profile in living cells (Fig. 2), and in postmortem brain tissue (Fig. 3) and behavioral responses (Fig. 4).

Molecular dynamics (MD) simulations have been shown to be a valuable tool for interrogating structural and dynamic events linked to GPCR function[32–34] including signaling bias[35]. Here, we exploited this approach to elucidate the structural determinants that may be responsible for the different in vitro and in vivo responses of the 5-HT$_{2A}$R. For this, we constructed 3D structural models of the complexes by docking all ligands into the orthosteric binding site of the 5-HT$_{2A}$R and subjected each complex to MD simulation. Structural inspection of the most representative clusters found in simulations reveals that the positioning of the main tryptamine scaffold of all compounds resembles the experimentally solved tryptamine pose of serotonin in the 5-HT$_{1A}$R (PDB id 7E2Y) with the following interactions of high contact frequencies (Fig. 5A, B): (i) a salt bridge with D3.32 and (ii) a hydrophobic sandwich formed of V3.33, F6.51 and F6.52. In fact, these positions have been corroborated by numerous mutational studies for tryptamine, 5-hydroxytryptamine and other closely related derivatives

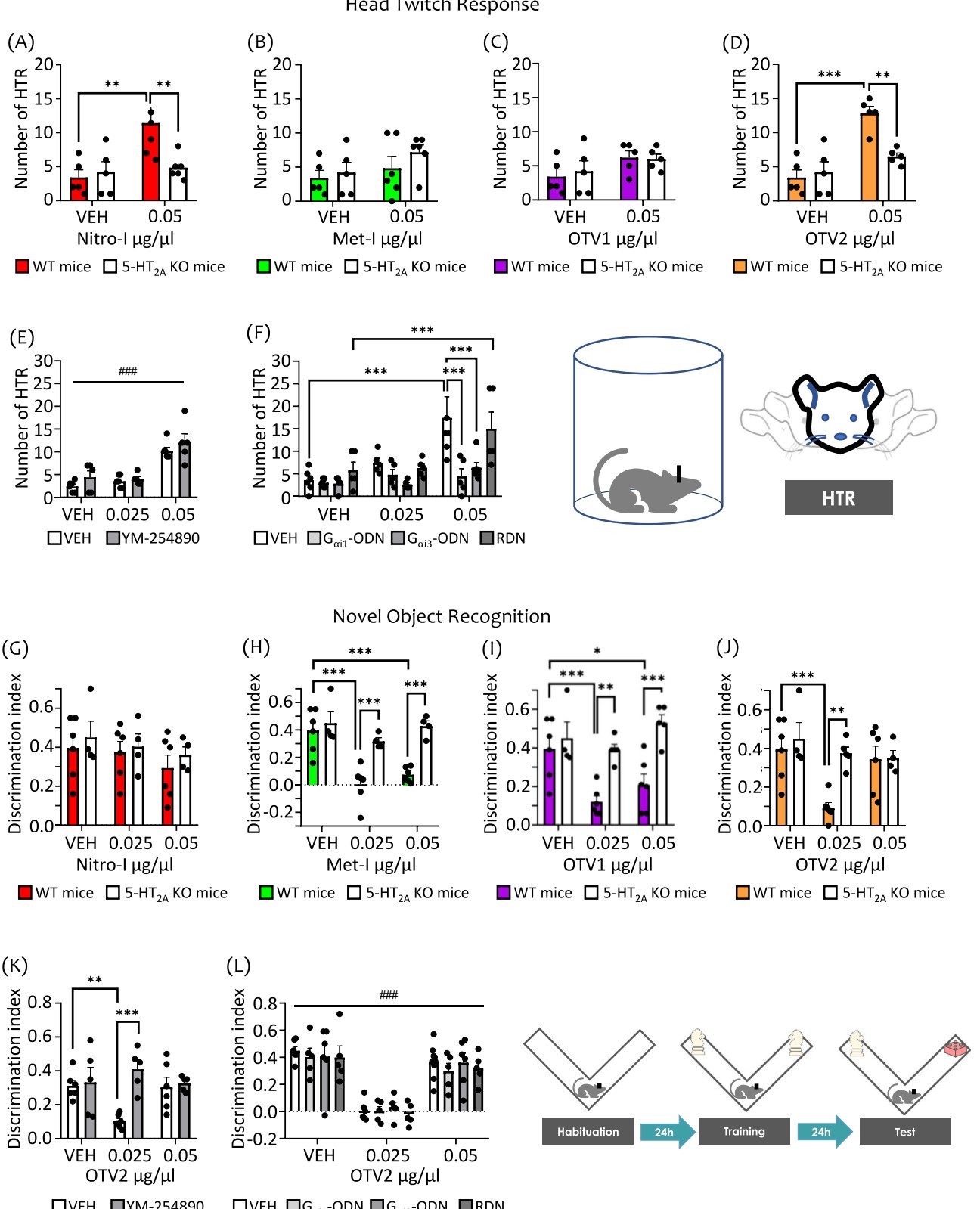

(i.e., V3.33[36], F6.51, and F6.52[37–39]). With this central scaffold in place, extension in position 5, as observed in OTV1 (1,4 benzodioxin) and OTV2 (phenoxy), are oriented towards the extracellular loop 2 (ECL2) where they result in increased contact frequencies (Fig. 5C). This interaction is mediated by hydrophobic interactions, in particular with L228 and L229, but also by additional hydrogen bonds with the backbone of L229 in the case of OTV1. The role of ECL2, specifically L229, mediating interactions with bulkier compounds in serotonin receptors has been reported by Wacker et al.[40].

Interestingly, we observe that differential ligand-receptor interactions are associated with distinct ligand binding affinities at the 5-HT$_{2A}$R in cell-based assays (Fig. 5D). The highest affinity for 5-HT$_{2A}$R

**Fig. 4 | Head twitch response (HTR) and long-term novel object recognition (NOR).** HTR in wild-type (WT) and 5-HT$_{2A}$R knockout (KO) mice following ICV administration of (**A**) Nitro-I, (**B**) Met-I, (**C**) OTV1, and (**D**) OTV2, or vehicle (VEH). The increase in HTR induced by Nitro-I and OTV2 at the dose of 0.05 μg/μl was absent in KO mice. **E** ICV administration of YM-254890 (16 μM) did not modulate the increase in HTR induced by OTV2 at the dose of 0.05 μg/μl. **F** ICV administration of specific antisense oligonucleotides (ODN), G$_{\alpha i1}$-ODN, G$_{\alpha i3}$-ODN, but not control random oligonucleotides (RDN) blocked HTR induced by OTV2 at the dose of 0.05 μg/μl. NOR memory test in WT and 5-HT$_{2A}$R KO mice following ICV administration of (**G**) Nitro-I, (**H**) Met-I, (**I**) OTV1, and (**J**) OTV2 or vehicle (VEH). In WT mice,

Met-I and OTV1 induced memory deficits at the dose of 0.025 and 0.05 μg/μl, and OTV2 was effective only at the dose of 0.025 μg/μl. These effects were abrogated in KO mice. Nitro-I did not trigger memory deficits at any of the doses tested. **K** YM-254890 (16 μM) abrogated the memory deficits induced by OTV2 at the dose of 0.025 μg/μl. **L** ICV administration of G$_{\alpha i1}$-ODN, G$_{\alpha i3}$-ODN, or the control RDN sequence did not modulate the memory deficits induced by OTV2 at the dose of 0.025 μg/μl. The data represent mean ± SEM. The number of mice used in the experiments (n) corresponds to the individual points in the graph. **p < 0.01, ***p < 0.001, and ###p < 0.001 (main effect of treatment). Two-way ANOVAs followed by Fisher's post-hoc test.

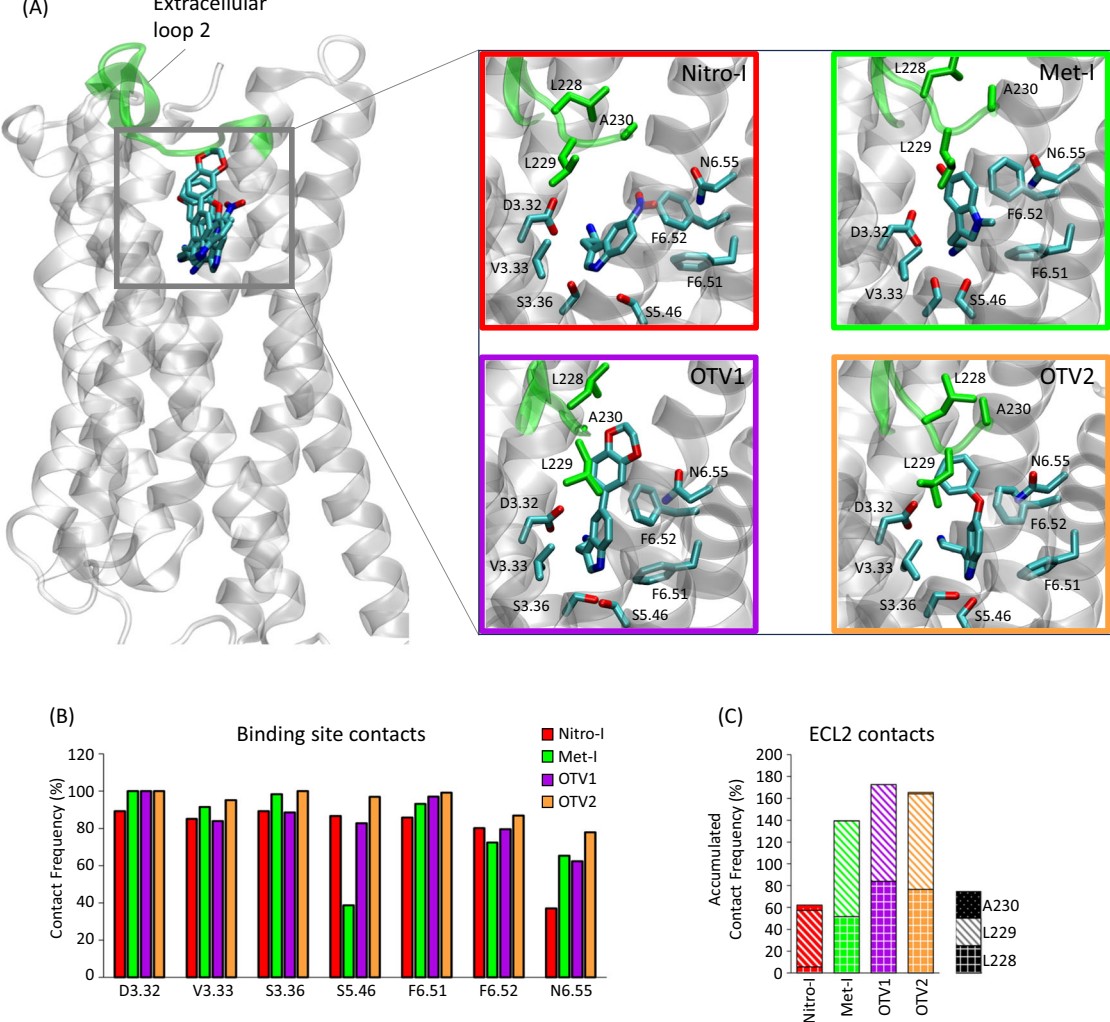

**Fig. 5 | In silico modeling of the ligand-5-HT$_{2A}$R interactions for Nitro-I, Met-I, OTV1, and OTV2. A** 3D model of the ligand-5-HT$_{2A}$R interactions for Nitro-I, Met-I, OTV1, and OTV2. The ECL2 and included residues are highlighted in green. **B** The contact frequency (%) of residues in the transmembrane region of the receptor for each ligand. Contact frequencies have been computed for the

main conformational cluster (see Methods) extracted from three replicates of 500 ns MD simulations (3 × 500 ns). **C** Accumulated contact frequency of residues in the ECL2 for each ligand computed for the main cluster extracted from three replicates of 500 ns MD simulations (3 × 500 ns).

is found for OTV2 which exposes a phenoxy substituent in position 5 of the indole fragment. Elongation of this position to a benzodioxan dramatically reduces ligand binding affinity as seen for OTV1. In addition, we observe that introducing a methyl substituent at the heteroaromatic nitrogen in position 1 is not favorable for ligand binding affinity, as observed for Met-I. One could speculate that ligand binding affinities are correlated with the total amount of receptor contacts (i.e. pocket plus ECL2 contacts). However, this is not the case as specific contact types (e.g. hydrogen bonds or Van der Waals)

contribute differently to the binding affinity which is not taken into account when computing the total number of contacts.

Nevertheless, the type of substitution in position 5 of the indole scaffold and corresponding ECL2 contacts seem to alter 5-HT$_{2A}$R coupling in living cells (Fig. 2). Compounds with small-sized substitutions in position 5 (i.e., Nitro-I and Met-I) show overall G$_{\alpha q}$ physiology-bias (compared to 5-HT) over the tested G$_{\alpha i}$ proteins as well as β-arrestin 1 and 2 (Fig. 2H). Changes are observed upon extension of position 5. Both, OTV1 (5-benzodioxin substituent) and OTV2

(5-phenoxy substituent) show a $G_{\alpha i1}$ bias over the canonical $G_{\alpha q}$ protein (Fig. 2H). Our structural models of ligand binding suggest that this change in coupling profile could be related to increased interaction with the ECL2 via the extended substituent in position 5 (Fig. 5C).

Structural modifications of the indole scaffold also significantly impact the 5-HT$_{2A}$R-induced G protein activation profile in human brain tissue: Nitro-I ($G_{\alpha i1}$, $G_{\alpha q/11}$ agonism), Met-I (inverse $G_{\alpha i1}$ agonism and $G_{\alpha i3}$, $G_{\alpha q/11}$ agonism), OTV1 ($G_{\alpha i1}$, $G_{\alpha i3}$, $G_{\alpha q/11}$ agonism) and OTV2 (inverse $G_{\alpha i1}$ agonism and $G_{\alpha i3}$, $G_{\alpha q/11}$ agonism). For instance, our data suggest that increased ECL2 contacts (Met-I, OTV1, OTV2, Fig. 5C) promote $G_{\alpha i3}$ activation in brain tissue (Fig. 3C–E) compared to Nitro-I (Fig. 3B). One of the most important structural observations is that the regulation of the psychosis-associated $G_{\alpha i1}$ pathway can be mediated by diverse mechanisms. On one hand, we find that differential interaction within the ECL2 (Fig. 5A, C) can convert a $G_{\alpha i1}$ agonism of OTV2 (hydrophobic ECL2 interaction via 5-phenoxy) (Fig. 3E) into a $G_{\alpha i1}$ inverse agonism as observed for OTV1 (polar ECL2 interaction via 5-benzodioxan) (Fig. 3D). In addition, we find that $G_{\alpha i1}$ agonism can also be induced by compounds with a relatively low number of ECL2 contacts (Nitro-I, Figs. 3B, 5C).

Based on this finding and our structural models, we propose that $G_{\alpha i1}$ agonism does not require strong ECL2 contacts, whereas $G_{\alpha i1}$ inverse agonism can be induced via differential interactions with the ECL2. In fact, this is in line with structural observations for the hallucinogenic compound LSD (PDB 6WGT) that stimulates $G_{\alpha i/o}$ coupling[9] compared to the non-hallucinogenic compound lisuride (PDB 7WC7) with no $G_{\alpha i/o}$ coupling through 5-HT$_{2A}$R[9]. Both compounds are structurally closely related with a highly similar binding mode but show differences in their interaction frequencies with the ECL2 in MD simulations (Fig. S6). Finally, another relevant structural observation is that compounds that induce cognitive deficits (Met-I, OTV1 and OTV2) are characterized by pronounced contacts with the ECL2 when compared to Nitro-I.

Overall, our structural insights can have important implications for the rational design of drug candidates with a tailored signaling profile and in vivo response applied to the treatment of psychiatric diseases.

## Discussion

In this study, we have investigated the complex coupling profile of the 5-HT$_{2A}$R in living cells and in postmortem brain tissue. Furthermore, in mouse models we tested the ability of several 5-HT$_{2A}$R agonists to modulate behaviors that have been associated with SCZ, including psychosis-related effects, anhedonia and cognitive deficits. For this, we used small molecular probes closely related to the endogenous agonist 5-HT and monitored their impact on receptor coupling preferences. The potential of such a strategy has been previously demonstrated by us for exploring the coupling bias of the dopamine D$_2$ receptor[35]. Exploiting this framework, we were able to detect structural determinants at the level of receptor binding that could relate to specific 5-HT$_{2A}$R-induced responses. Specifically, we find that the degree of ligand interaction with the ECL2 has a dramatic impact on the receptor's potency and efficacy to couple to different G protein subtypes in living cells (Fig. 2). This in turn translates into specific physiology-bias profiles for different coupling pathways. For instance, small 5-HT-like compounds (Nitro-I, Met-I) show an overall $G_{\alpha q}$ bias over the $G_{\alpha i}$ family and β-arrestins 1 and 2 (Fig. 2H), whereas extensions in position 5 (OTV2) can convert the coupling preference to the $G_{\alpha i}$ family over the $G_{\alpha q}$ in living cells (Fig. 2H). A specific property of this molecular extension is to promote increased interactions with ECL2 (Fig. 5C) which is potentially responsible for an altered signaling response. In fact, this is in line with a study from Wacker et al. [40], showing that mutational modifications in the ECL2 of the 5-HT$_{2B}$R drive ligand binding kinetics and receptor response in cell-based assays[40]. Of note, our study goes beyond cell-based responses and also interrogates the ligand-mediated impact on

signaling in native brain tissue. Our data suggest that one possible mechanism for modulating agonism (OTV2, Fig. 3E) or inverse agonism (OTV1, Fig. 3D) of the $G_{\alpha i1}$ pathway involves differential interactions with the ECL2 (Fig. 5C). Interestingly, this finding goes along with structural observations for the hallucinogen LSD ($G_{\alpha i/o}$ stimulation) and the non-hallucinogenic lisuride (no $G_{\alpha i/o}$ stimulation)[9,41–43]. Experimentally solved structures of these two closely related compounds bound to the 5-HT$_{2A}$R show main differences in ECL2 interactions that could be responsible for the differential $G_{\alpha i/o}$ family coupling properties (Fig. S6). In contrast to this, ECL2 contacts seem to be of less relevance for $G_{\alpha i1}$ agonism, as demonstrated by Nitro-I, which shows a marked reduction of ECL2 contacts. The obtained structural insights are of high importance for the tailored design of compounds with a specific coupling profile. Of note, our structural observations are limited to compounds with a 5-HT scaffold. Taking into account the complexity of signaling responses, we cannot exclude the existence of additional mechanisms that can drive the observed effects.

Ultimately, to examine the impact of different 5-HT$_{2A}$R signaling profiles on behavioral responses, we moved to in vivo experiments evaluating psychosis-related effects, cognitive deficits, and depression-like behavior induced by our probes[12,20–24,44].

Interestingly, the present results demonstrate that compounds that elicit HTR (Nitro-I and OTV2, Fig. 4A, D) are able to induce activation of the $G_{\alpha i1}$ subunit (Fig. 3B, E), while compounds that did not produce HTR (Met-I and OTV1, Fig. 4B, C) show inverse agonism towards $G_{\alpha i1}$ in both human (Fig. 3C, D) and mouse brains (Fig. 3G, H). Importantly, the implication of $G_{\alpha i1}$ in HTR (and thus psychosis-related effects) is further corroborated by our finding that a reduction of the expression levels of $G_{\alpha i1}$ together with $G_{\alpha i3}$ reversed 5-HT$_{2A}$R-mediated HTR (Fig. 4F). Although no discrimination between the roles of $G_{\alpha i1}$ and $G_{\alpha i3}$ could be made in this experiment, our data suggest that $G_{\alpha i1}$ activation is driving psychosis-related effects as found in Nitro-I and OTV2 based on: (i) only compounds that promote HTR (Nitro-I and OTV2) activate $G_{\alpha i1}$ while others do not (Met-I and OTV1) and (ii) all studied compounds (Nitro-I, Met-I, OTV1, OTV2) activate $G_{\alpha i3}$ independently of their ability to induce/not induce pro-psychotic effects suggesting a marginal implication of $G_{\alpha i3}$ in pro-psychotic effects. Future studies with selective inhibitory tools are required to further investigate the specific implication of $G_{\alpha i1}$ in the psychosis-related effects. It is worth noting that our observation is in agreement with previous findings suggesting that both pro-hallucinogenic and anti-hallucinogenic properties of 5-HT$_{2A}$R drugs depend on the modulation of $G_{\alpha i/o}$ proteins and their downstream pathways. However, these studies do not further differentiate the precise G protein subunits involved in these processes[9,45].

Altogether, our findings indicate that inhibition of 5-HT$_{2A}$R-mediated $G_{\alpha i1}$ activity could be a promising strategy to selectively reduce pro-psychotic symptoms. Moreover, this approach could down-regulate the supersensitivity of the 5-HT$_{2A}$R coupling to $G_{\alpha i1}$-proteins (but not to $G_{\alpha q/11}$), which has been reported to occur in postmortem brains of subjects with schizophrenia[41]. On the other hand, Met-I, Nitro-I and OTV2 exhibit similar efficacy/potency for recruiting β-arrestin 1 and 2 in BRET assays, but only Met-I and OTV2 elicited HTR in mice. These findings suggest that activation of β-arrestin may be necessary, but not sufficient for the appearance of HTR (see supplemental section 2).

The 5-HT$_{2A}$R is also a key player in 5-HT's regulation of learning and memory[44,46]. Previous studies have investigated the effect of structurally diverse 5-HT$_{2A}$R ligands in these processes (see supplemental section 3). However, to our knowledge, no study has investigated the implication of specific 5-HT$_{2A}$R-induced pathways in cognition. Interestingly, examining our 5-HT$_{2A}$R signaling probes, we find that Met-I, OTV1 and OTV2 but not Nitro-I induced 5-HT$_{2A}$R mediated long-term memory deficits in WT mice (Fig. 4G–J), mirroring the alterations in cognitive processing in SCZ. Most importantly, our experiments provide evidence that $G_{\alpha q}$ is a critical element in mediating cognitive

processes via the 5-HT$_{2A}$R, as its pharmacological inhibition reversed the observed long-term memory alterations in mice (Fig. 4K).

Ultimately, our study highlights the challenges in multidisciplinary research related to the fact that data are often not completely comparable between different experimental setups. Recognizing this divergence in data becomes paramount, as it not only informs researchers about the existing challenge but also serves as a means to elucidate the extent of in vitro, in vivo and human data alignment. In addition, the differences observed between these setups can have significant implications for gaining a better understanding of the disease mechanisms underlying schizophrenia (e.g. diverse expression levels of coupling partners, etc.).

In conclusion, widely used atypical antipsychotics usually target hallucinations and paranoid thinking, but their beneficial effects on cognitive symptoms are controversial[47]. In addition, current treatments can induce severe side effects which are potentially the result of indiscriminate inhibition/activation of several pathways that can be initiated by one receptor target. Our findings highlight that 5-HT$_{2A}$R pathway-biased antagonists/inverse agonists that selectively target G$_{\alpha i}$ pathways, and specifically the G$_{\alpha i1}$, could improve positive symptoms without affecting cognitive processing. In contrast, drugs that selectively block G$_{\alpha q/11}$ signaling could be good therapeutic agents for patients that suffer from cognitive disturbances. Importantly, our work provides structural insights into ligand-receptor interactions, which are of high relevance for the rational design of drugs with desired therapeutic profiles. Beyond this, our work highlights the complexity of GPCR signaling and the relevance of G protein-specific mechanisms for the therapeutic response, which has to be considered for future drug development efforts of more efficient and safer drugs in the treatment of psychiatric diseases.

## Methods

### Ethical statement
Animal procedures were carried out following the standard ethical guidelines (European Communities Directive 86/609 EEC) and approved by the local ethical committee (CEEA-PRBB).

Human brain samples were obtained at autopsy in the Basque Institute of Legal Medicine, Bilbao, Spain, in compliance with Spanish policies of research and ethical boards for postmortem brain studies. According to applicable laws, samples were obtained by opting-out policy and absence of compensation for tissue donation. The Project was approved by the Institutional Review Board (IRB) of the University of the Basque Country UPV/EHU (CEISH-UPV/EHU, Ref. M10-2019-230).

### Virtual screen for 5-HT$_{2A}$R ligands
Four virtual screens were conducted with the ZINC database[48] of commercially available 'lead-like' compounds at the orthosteric binding site of 5-HT$_{2A}$R using four different conformational ensembles derived from the previous publication[3]. The Glide module in Schrödinger was employed to implement the hierarchy screening workflow[49]. After completing the first step in the quick HTVS mode, the top 100,000 solutions were chosen for the next screen using the Standard Precision (SP) procedure, which uses a more precise scoring function and slower but thorough ligand conformational sampling. To guarantee that the important ligand-protein interactions were captured, the H-bond constraints to D3.32, N6.55, and S5.46 were applied. The most promising compounds were then chosen after visually evaluating the remaining top-ranked compounds based on the interactions between ligands and proteins, shape complementarity, lead-like characteristics, lack of potentially reactive and PAINS groups, and chemical diversity. The experimental validation led to the identification of the molecules OTV1 and OTV2 - two structurally related compounds of Met-I and Nitro-I, which emerged from our previous study[3].

### Drugs, antibodies, and reagents
The following ligands were used: 2-[5-(2,3-dihydro-1,4-benzodioxin-6-yl)-1H-indol-3-yl]ethan-1-amine Hydrochloride (Otava 3575001; OTV1) and 2-(5-phenoxy-1H-indol-3-yl)ethan-1-amine hydrochloride (Otava 3736689; OTV2) from Otava Chemicals. 2-(5-nitro-1H-indol-3-yl)ethamine hydrochloride (Nitro-I), 3-(2-aminoethyl)-1-methyl-1H-indol-5-ol hydrochloride (Met-I) and (±)-2,5-dimethoxy-4-iodoamphetamine hydrochloride (DOI HCl) from Sigma-Aldrich Merck. α-phenyl-1-(2-phenylethyl) −4-piperidinemethanol (MDL-11,939) from Bioscience (UK). MDL-11,939 was chosen as a selective 5-HT$_{2A}$R *vs* 5-HT2CR antagonist[50,51]. [$^{35}$S]GTPγS was purchased from Perkin Elmer Life Sciences (Boston, USA). Other reagents for SPA were obtained from Sigma-Aldrich and Perkin Elmer Life Sciences. The antibodies used for in vitro functional assays are further described in Supplementary Section 4 and in Table S2.

In behavioral experiments, Nitro-I, Met-I, OTV1, and OTV2 (0.025, 0.05, 0.1, and 0.5, μg/μl) were diluted in 99% saline with 1% DMSO (Sigma-Aldrich Merck), (±)DOI (0.1 μg/μl) was diluted in saline. These compounds were administered intracerebroventricularly (ICV) at a volume of 5 μl. The G$_{\alpha q/11}$ inhibitor YM-254890 (YM, FUJIFILM Wako Pure Chemical Co) was reconstituted with DMSO to provide a stock solution of 1 mM. The stock solution was diluted with saline to a final concentration of 16 μM and administered ICV at a volume of 2.5 μl. The antisense oligodeoxynucleotides (ODNs) that inhibit G$_{\alpha i1}$ and G$_{\alpha i3}$ and a random oligo (ODN-RD) that served as a control (Sigma-Aldrich Merck) were reconstituted with Milli-Q water to provide a stock solution in the appropriate concentrations for the ICV administration of the different doses at a volume of 2.5 μl. Sequences were as follows: ODN-G$_{\alpha i1}$: 5′-G*C*TGTCCTTCCACAGTCTCTTTATGACGCCG*G*C-3′, corresponding to nucleotides 588–621 of the GNAI1 gene sequence. ODN-G$_{\alpha i3}$: 5′-G*C*CATCTCGCCATAAACGTTTAATCACGCCT*G*C-3′, corresponding to nucleotides 554–587 of the GNAI3 gene sequence. These sequences showed no homology to other relevant cloned proteins (GeneBank database). ODN-RD with the sequence 5′-C*C*CTTATTTAC-TACTTTC*G*C-3′[26].

### Bioluminescence resonance energy transfer assay (BRET)
To measure activation of the different G protein pathways and to detect recruitment of the β-arrestins, enhanced bystander BRET (ebBRET) Effector Membrane Translocation Assay (EMTA) biosensors[9] were used in HEK-293 cells. HEK-293 clonal cell lines were a gift from S. Laporte (McGill University, Montreal, Quebec, Canada). The plasmids p63-RhoGEF-RlucII, Rap1Gap-RlucII, β-arrestin1-RlucII, β-arrestin2-RlucII and rGFP-CAAX have been previously described[52–54] and the human 5-HT$_{2A}$R was a gift from Domain Therapeutics North America. See Supp. information for more details.

### Ligand binding studies
A total of 5-HT$_{2A}$R competition binding experiments were carried out in a polypropylene 96-well plate. In each well was incubated 70 μg of membranes from CHO-5-HT 2 A cell line prepared in our laboratory (Lot: A006/10-03-2020, protein concentration=5134 μg/ml), 1 nM [$^3$H]ketanserin (47.3 Ci/mmol, 1 mCi/ml, Perkin Elmer NET791250UC) and compounds studied and standard. Non-specific binding was determined in the presence of methysergide 1 μM (Sigma M137). The reaction mixture (Vt: 250 μl/well) was incubated at 37 °C for 30 min, 200 μl was transferred to GF/B 96-well plate (Millipore, Madrid, Spain) pretreated with 0.5% of PEI and treated with binding buffer (Tris-HCl 50 mM, pH=7.4), after was filtered and washed six times with 250 μl wash buffer (Tris-HCl 50 mM, pH=6.6), before measuring in a microplate beta scintillation counter (Microbeta Trilux, PerkinElmer, Madrid, Spain).

### Brain prefrontal cortex membranes preparation
Human brain samples were obtained at autopsy in the Basque Institute of Legal Medicine, Bilbao, Spain, in compliance with policies of research

and ethical boards for postmortem brain studies at the moment of sample obtaining. Thus, samples from 12 different subjects (10 males and 2 females) with ages between 29–90 years were included. The postmortem delay between death and storage of the samples ranged from 4 to 12 h, and the storage time between sampling and experiments ranged from 48 to 10 months. All the subjects were determined to be free of neurological and psychiatric disorders based on medical records and postmortem tissue examinations. Positive blood toxicology for drugs or ethanol was considered exclusion criteria. Samples from the dorsolateral prefrontal cortex (PFC) were dissected at autopsy following established protocols[55] and immediately stored at −70 °C until assay. Adult C57BL/6 J mice were sacrificed by cervical dislocation, brains removed, PFC samples dissected, and samples stored at −70 °C until assay. See Supp. Information for more details.

### Antibody-capture [$^{35}$S]GTPγS binding scintillation proximity assay (SPA)

Specific activation of different subtypes of G proteins was determined using a homogeneous protocol of [$^{35}$S]GTPγS SPA coupled with the use of specific antibodies essentially as previously described[16,17] following experimental conditions for determination of agonism, antagonism or inverse agonism properties of tested drugs. A single submaximal concentration of the drugs (10 μM) was used. This submaximal concentration was chosen, as previously reported[18,56,57], as able to give us binding values around the maximal effect for any drug and subunit subtype combination studied. See Supplementary Information for more details on antibody specificities (Fig. S8).

### Behavioral studies

**Animals.** We used male homozygous 5-HT$_{2A}$R KO mice and WT littermates on a C57BL/6 J background (Charles River, France) (25-30 g). Since females have not been tested, this study does not address sex-related differences. 5-HT$_{2A}$R KO mice and WT littermates were bred in the animal facilities of the PRBB. Mice were initially grouped-housed in a room with controlled-temperature (21 ± 1 °C) and humidity (55 ± 10%) environment with food and water available *ad libitum*. All the experiments were performed during the dark phase of the light/dark cycle (lights off at 8 a.m. and on at 8 p.m.), by observers blind to the experimental conditions.

**Intracerebroventricular surgery and infusion.** Nitro-I, Met-I, OTV1 and OTV2 and DOI were administered intracerebroventricularly (ICV). See Supp. Information for more details.

### Behavioral Tests

**Irwin test, head twitch response (HTR).** Immediately following the ICV infusion, mice were placed in a transparent Plexiglas cylinder (30 cm in diameter and 50 cm high) and the Irwin test and HTR were quantified during 30 min. In a modified Irwin test[58] we evaluated behavioral dysfunction produced by 5-HT$_{2A}$R ligands and to estimate the minimum lethal dose and the range dose of each compound (data not shown). Some symptoms were evaluated by their presence or absence (lethality, convulsions, straub tail, abnormal gait, jumps, motor incoordination, piloerection, tremor, excitation, low reactivity to touch, and akinesia). Other symptoms, such as stereotypic behaviors (grooming, rearing, and scratching), were measured by the sum of events that occurred in 30 min. The HTR, characterized by a rapid side-to-side rotational head movement, was measured during 30 min following drug administration[59].

**The novel object recognition (NOR) test.** This test was performed to evaluate long-term memory deficits, as previously described[19]. The test consists of a black Plexiglas "V" maze with two corridors (30 cm long x 4.5 cm wide, and 15 cm high walls) set at a 90° angle (Panlab, Barcelona, Spain). See Supp. Information for more details.

**Sucrose preference test.** The sucrose preference test was used to evaluate negative symptoms associated with schizophrenia, such as anhedonia[60]. See Supp. Information for more details.

**Experimental procedures.** To examine the dose-related effects of the compounds on HTR, different doses of pharmacological probes (0.025, 0.05, 0.1 and 0.5 μg/μl) or vehicle (VEH) were administered to C57BL/6 J. A set of mice received all doses of OTV1, and OTV2 and VEH (n = 5–11). Another set of mice received all doses of Nitro-I and VEH (*n* = 6–11), and an additional set of mice received all doses of Met-I (0.025, 0.05, 0.1, and 0.5 μg/μl) and VEH (n = 5-12) in a Latin square design with a 3-day wash-out period between ICV infusions. Immediately after ICV infusions, mice were placed in a transparent Plexiglas cylinder and HTR were quantified during 30 min.

To evaluate the role of 5-HT$_{2A}$R on the behavioral responses to the different compounds, WT (n = 5) and 5-HT$_{2A}$R KO (n = 6) mice received OTV1, OTV2, Nitro-I, Met-I (0.025, 0.05 μg/μl) and VEH in a Latin square design with a 3-day wash-out period between ICV infusions. Mice were first habituated (day 1) and then trained (day 2) in the NOR apparatus. Immediately after training, mice received ICV infusions and were placed in a transparent Plexiglas cylinder for 30 min to evaluate the Irwin test and HTR. 24 h after the ICV infusion, mice were tested in the NOR apparatus. Following a 3-day wash-out period, mice were habituated to the sucrose preference test for 3 days. On the fourth day, mice again received ICV infusions, and were presented with two volumetric pipettes, one containing drinking water and the other containing 1% sucrose. The intake of water and sucrose was measured daily every 24 h during 3 days.

To evaluate the role of G$_{αq/11}$ on HTR and long-term memory deficits induced by OTV2 (0.025, 0.05 μg/μl), two separate Latin square designs were implemented in naive C57BL/6 J mice. Mice were habituated (day 1) and trained (day 2) to the NOR apparatus. Immediately after training, they received an ICV infusion of a vehicle solution, and 220 min later, ICV infusions of OTV2 were performed and HTR was measured. The following day, the NOR test was performed. Another group (*n* = 5) followed the same procedure but instead first received one ICV infusion of the G$_{αq/11}$ inhibitor YM-254890 (16 μM), and 220 min later, ICV infusions of OTV2 were performed (Fig. S7A).

To evaluate the role of G$_{αi1}$ and G$_{αi3}$ on the effects of OTV2, four sets of C57BL/6 J mice (n = 5/group) were tested according to a Latin square design with a 3-day wash-out period in between infusions. One set received ICV infusions of either ODN, ODN-RD or distilled water as vehicle using the following schedule: on days 1 and 2 mice received 1 nmol, on days 3 and 4 they received 2 nmols, and on day 5 they received 3 nmols, as previously described[61]. On day 5 (before the ICV infusions), mice were habituated in the NOR. On day 6, mice were first trained in the NOR and then received ICV infusions of OTV2, and the HTR was measured for 30 min. On day 7, the NOR test was performed (Fig. S7B).

### Western blots

To test whether G$_{αi1}$ and G$_{αi3}$ protein levels were significantly decreased by specific ODNs, Western blot analysis was carried out (see Supp. Information for more details).

### Structural models for molecular dynamics simulations

The receptor was modeled based on PDB ID 6WHA, which is an active structure bound to an agonist, 25CN-NBOH, and coupled to miniG$_q$. To curate the structure, thermostabilizing and homogenizing alanine mutations at L247 and L371 were reverted. The missing parts of the structure (the beginning of the TM1, extracellular loops and missing side chains) were modeled using MODELLER[62]. Of note, multiple structures (PDB ID 6A93 and PDB ID 6A94 structures) were used as templates for modeling the extracellular loop. The four small molecules, including Nitro-I, Met-I, as well as the newly retrieved OTV1 and

OTV2, were docked to the receptor using the Molecular Operating Environment (MOE) with the triangle matcher as the placement method. The program was set to generate 30 poses for each molecule. The top 5 remaining after visual inspection were subjected to energy minimization using the MMFF94x force field while keeping the receptor rigid. The resulting poses were further analyzed, with special attention to the interaction with D3.32, as this is necessary for the activation of the aminergic receptors.

**Molecular dynamics simulations and conformational clustering**
In a next step, systems of the obtained ligand-5-HT$_{2A}$R complexes embedded in a hydrated membrane bilayer were generated. For this, internal waters were introduced into the receptor using Homolwat[63], and the protonation states of protein residues were decided using ProteinPrepare from the python module HTMD[64]. In a next step, the ligand receptor complexes were embedded in a lipidic POPC bilayer, solvated with TIP3 waters and ionized with 150 mM Na+ or Cl$^-$ ions. Models were equilibrated using ACEMD3[65] in NPT conditions for 40 ns. To ensure sufficient sampling of ligand-receptor contacts, each system was simulated for $3 \times 500$ ns (1.5 µs per system) in NVT conditions using ACEMD3 according to the state-of-the-art[34]. Non-bonded interactions were cut-off at 9 Å. A smooth switching function for the cut-off was applied, starting at 7.5 Å. Long-distance electrostatic forces were calculated using the Particle Mesh Ewald algorithm. All simulations were carried out at a temperature of 310 K using the Langevin thermostat with damping constants γ of 1 ps$^{-1}$ and 0.1 ps$^{-1}$ for NPT and NVT simulations, respectively.

The trajectories obtained from the three replicates were concatenated into one trajectory file and clustered based on the movements of ligands using the average linkage analysis in CPPTRAJ[66]. The cut off for the RMSD of the ligand for clustering was set to 5 Å. The contact and binding mode analysis were performed on the main cluster of each system. Receptor-ligand contacts were investigated in the GPCRmd workbench (www.gpcrmd.org). Contacts within 3 Å of the ligand which occurred with a frequency above 50% in the main cluster of the MD simulations were investigated.

The simulation data are made available via the GPCRmd repository (www.gpcrmd.org)[34]:

Met-I bound to 5-HT$_{2A}$R: https://submission.gpcrmd.org/view/1105/
Nitro-I bound to 5-HT$_{2A}$R: https://submission.gpcrmd.org/view/1107/
OTV1 bound to 5-HT$_{2A}$R: https://submission.gpcrmd.org/view/1128/
OTV2 bound to 5-HT$_{2A}$R: https://submission.gpcrmd.org/view/1110/
LSD bound to 5-HT$_{2A}$R (PDB 6WGT): https://submission.gpcrmd.org/view/1175/
Lisuride bound to 5-HT$_{2A}$R (PDB 7WC7): https://submission.gpcrmd.org/view/1176/

**Statistical analysis**
In dose-response studies, one-way ANOVA were used to analyze the behavioral data comparing the different doses of Nitro-I, Met-I, OTV1 or OTV2. An unpaired Student's t test was used to analyze the data for DOI versus VEH administration. In studies evaluating the role of 5-HT$_{2A}$R, two-way ANOVAs were used to analyze the behavioral data with genotype (WT and 5-HT$_{2A}$R KO mice) as a between subject factor, and treatment (different doses of Nitro-I, Met-I, OTV1 or OTV2 and VEH) as a within subject factor. For G-protein studies, two-way ANOVAs were used to analyze the behavioral data with G-protein inhibitors (YM-254890, G$_{\alpha i1}$-ODN, G$_{\alpha i3}$-ODN, ODN-RD and VEH) as a between subject factor, and treatment (different doses of OTV2 and VEH) as a within subject factor. Fisher's LSD *post-hoc* tests for multiple comparisons were performed when appropriate. Specific [$^{35}$S]GTPγS binding results were analyzed by one- and two-sample Student's *t*-test *vs* basal values (expressed as 100%) or between experimental groups, respectively. For Western blot assay,

statistical analysis for comparison of the means between random and specific ODN treatments was performed by a one-way ANOVA followed by *post hoc* Bonferroni for multiple comparisons. The statistical analyzes were performed with the "Statistica" programme, version 6 (StatSoft Inc.) and GraphPad Prism™. All data are presented as mean ± SEM and statistical significance was set at p < 0.05 level. The statistical values obtained for all behavioral studies are shown in Table S3.

**Reporting summary**
Further information on research design is available in the Nature Portfolio Reporting Summary linked to this article.

## Data availability
Data supporting the findings of this manuscript are available as a Supplementary Information file. MD simulations are deposited at the GPCRmd database (www.gpcrmd.org). Met-I bound to 5-HT$_{2A}$R: [https://submission.gpcrmd.org/view/1105/]. Nitro-I bound to 5-HT$_{2A}$R: [https://submission.gpcrmd.org/view/1107/]. OTV1 bound to 5-HT$_{2A}$R: [https://submission.gpcrmd.org/view/1128/]. OTV2 bound to 5-HT$_{2A}$R: [https://submission.gpcrmd.org/view/1110/]. LSD bound to 5-HT$_{2A}$R (PDB 6WGT): [https://submission.gpcrmd.org/view/1175/]. Lisuride bound to 5-HT$_{2A}$R (PDB 7WC7): [https://submission.gpcrmd.org/view/1176/]. Additional data supporting the findings are available from the corresponding authors upon request. A Source Data file is included with this manuscript. Source data are provided with this paper.

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

## Acknowledgements

This work was supported by the ERAnet NEURON consortium fund (funding was provided by CIHR NDD-161471 and FRQ-S 278647 for M.B., the German Federal Ministry of Education and Research under grant number 01EW1909 for P.K., as well as the Instituto de Salud Carlos III and Fondo Europeo de Desarrollo Regional number AC18/00030 for J.S. and P.R.). This work was further supported by the Instituto de Salud Carlos III (ISCIII) and co-funded by the European Union (PI18/00094) to J.S. and (PI18/00053) to P.R. We acknowledge grant support from Agencia Estatal de Investigación (PID2020-119428RB-I00; SAF2017-88126R), Basque Government (IT-1211/19, IT-1512/22 and KK-2019/00-49), Xunta de Galicia (ED431C 2022/20 and ED431G 2019/02) and European Regional Development Fund (ERDF). P.K. thanks the German Research Foundation DFG for Heisenberg Professorship KO4095/5-1. S.S.O. and R.S. thank the PTQ-17-09103 (Ayuda Torres Quevedo, Ministerio de Ciencia e Innovación), and BioExcel-2 (Grant Number 823830, Horizon2020). M.B. was in part supported by an operating grant (# PJT 183758) from the Canadian Institute for Health Research. I.M-A. was the recipient of a predoctoral fellowship from the Basque Government. The authors would like to thank the staff members of the Basque Institute of Legal Medicine for their cooperation in the study, especially to Dr. Benito Morentin. R.D-A., T.M.S, D.A.G., I.M.A., A.S., P.K and J.S. are members of COST Action CA18133 "ERNEST".

## Author contributions

P.R. and J.S. conceived and designed the study. E.K. and C.R.D. supervised by R.T. and P.R. performed the behavioral studies in mice. R.D.A., I.M.A., and J.J.M. carried out receptor coupling experiments in human and mice brain tissue. S.A.G. and E.T. supervised by MB conducted the cell-based BRET assays. A.S., D.A.G., and T.M.S. supervised by P.K. and J.S. built structural models and carried out molecular dynamics simulations. S.S.O. and R.S. performed the virtual screening. D.M., J.B., and M.I.L. conducted the ligand binding assays. P.R. and J.S. wrote the manuscript with main contributions from R.D.A., E.K. and M.B. All authors revised the manuscript. P.R. and J.S. supervised and coordinated the whole project.

## Competing interests

The authors declare no competing interests.

## Additional information

[1]Integrative Pharmacology and Systems Neuroscience Research Group, Hospital del Mar Research Institute, Barcelona, Spain. [2]Department of Pharmacology, University of the Basque Country/Euskal Herriko Unibertsitatea, Leioa, Bizkaia, Spain. [3]Centro de Investigación Biomédica en Red de Salud Mental CIBERSAM, Madrid, Spain. [4]Instituto de Investigación Sanitaria Biobizkaia, Barakaldo, Bizkaia, Spain. [5]Department of Biochemistry and Molecular Medicine, Institute for Research in Immunology and Cancer (IRIC), Université de Montréal, Montréal, Québec H3T 1J4, Canada. [6]Cell-type mechanisms in normal and pathological behaviour Research Group, IMIM-Hospital del Mar Medical Research Institute, Barcelona, Spain. [7]Research Programme on Biomedical Informatics (GRIB),

Hospital del Mar Research Institute, Barcelona, Spain. [8]InterAx Biotech AG, PARK InnovAARE, 5234 Villigen, Switzerland. [9]Department of Medicine and Life Sciences, Pompeu Fabra University, Barcelona, Spain. [10]NBD NOSTRUM BIODISCOVERY, Av. de Josep Tarradellas, 8-10, 3-2, 08029 Barcelona, Spain. [11]Pharmaceutical Chemistry, University of Marburg, Marbacher Weg 8, Marburg 35037, Germany. [12]Innopharma Drug Screening and Pharmacogenomics Platform. BioFarma research group. Center for Research in Molecular Medicine and Chronic Diseases (CiMUS). Department of Pharmacology, Pharmacy and Pharmaceutical Technology, University of Santiago de Compostela, Santiago de Compostela, Spain. [13]Health Research Institute of Santiago de Compostela (IDIS), University Hospital of Santiago de Compostela (SERGAS), Trav. Choupana s/n, 15706 Santiago de Compostela, Spain. [14]These authors contributed equally: Elk Kossatz, Rebeca Diez-Alarcia. ✉e-mail: probledo@imim.es; jana.selent@upf.edu

