## [Peer Review File · Nature Communications]

G protein-specific mechanisms in the serotonin 5-HT_{2A} receptor regulate psychosis-related effects and memory deficitsReviewer #1 (Remarks to the Author):

This is a new submission by Kossatz et al testing the effect of 5-HT_{2A} ligands on G protein coupling and b-arrestin recruitment in cells, as well as functional assays and behavioral phenotypes in mice. The authors proposed that psychedelics induced HTR via Gi proteins, whereas deficits in cognitive performance via Gq. The manuscript is very well written and results are interest since they provide important evidence in support of biased agonist via inhibitory Gi proteins as a key mechanism behind the hallucinogenic properties of psychedelic 5-HT_{2A} agonists.

Concerns:

- 1- Minor: Fig 2 panel – “dose response” should be changed to “concentration response” since this figure panel shows in vitro findings
- 2- Data in vitro are totally unnecessary and confusing since ligands show very different agonist/inverse agonist properties as compared to mouse and human brain samples
- 3- Minor: mouse experiments were conducted in male mice only.
- 4- Demographic data of the postmortem human brain samples are missing in the methods section (sex, age, cause of death etc...)
- 5- Specificity/selectivity of antibodies is usually questionable. How the authors know that Gi1, Gi2, and Gi3 antibodies are specific/selective for this Galpha subunits? Have these antibodies been validated in KO mice? Lack of specificity/selectivity may also explain findings in Fig S4 – for example, it could be that the antiGalpha1 or anti-Galpha3 antibodies are not specific, and therefore knocking down one of these two Galpha proteins leads to reduced immunoreactivity with any of the two antibodies.
- 6- The authors describe in the Results section that “Met-I and OTV1 elicit inverse agonism at Galphai1...” – This reviewer does not see inverse agonism with Met-I on Galphai1 in Fig 3C.
- 7- The description of HTR as a model of positive symptoms in schizophrenia patients is incorrect. HTR is a mouse behavioral proxy of human hallucinogenic potential – this is not modeling “hallucinations” in mice. This part of the text should be corrected.
- 8- Docking in Fig 5 is interesting, but without experimental validation (mutations of the predicted binding pocket, for example) these computer modeling findings are unconvincing.
- 9- Back to Fig 2, authors show functional properties but not ligand binding assays. Findings testing affinity (binding displacement curves) of [³H]ketanserin are needed to validate if Nitro-I, Met-I, OTV1 and OTV2 bind to the orthosteric binding pocket of the 5-HT_{2A}
- 10- Source of the 5-HT_{2A} plasmid (or stable cell line) is not mentioned in the methods section.

Reviewer #2 (Remarks to the Author):

In this work, an extensive analysis of the signaling profile of four small molecule agonists of the serotonin 5-HT_{2A} receptor is reported in vitro, in vivo, and ex vivo, followed by in silico rationalization. These molecules – chemically similar to the endogenous agonist serotonin – have been assessed in a cell-based (BRET) assay for their effectiveness in recruiting 12 different effector proteins (from Gαq, Gαi/o, and β-arrestin families), from which distinct functional bias profiles are reported (two as Gαq-biased, one as Gi/o-biased, and one low potency in all pathways). Then, the signaling profiles were

investigated in postmortem human brain tissue, where all compounds were found to be 5-HT_{2A}-G_q pathway agonists, but some were found 5-HT_{2A}-G_i inverse agonists while others were agonists in this pathway(s). Then, via experiments with mice (head-twitch response and novel object recognition), the authors conclude that G_{i/o} signaling mediated by the 5-HT_{2A} receptor promotes pro-psychotic (hallucinogenic) effects while 5-HT_{2A}-mediated G_q signaling enhances memory performance. Finally, these differences in the signaling profile and behavioral responses are linked to specific (and differential) ligand-receptor interactions, in particular to residues of the extracellular loop-2 (ECL2) via computational modeling (molecular dynamics simulations).

I recognize the extensive multidisciplinary effort and the relevance of this work to advance our understanding of the physiopathological roles of the 5-HT_{2A} receptor. In particular, the roles that the multiple signaling pathways elicited by this receptor possess in the behavioral outcomes and its implication, e.g., to the understanding and treatment of schizophrenia and other mental illnesses. However, I have a few concerns about the study that I would like the authors to address before this manuscript can be considered for publication at Nature Communications. I list them below.

1) Very little attention is given to the fact that two of the four tested molecules are relatively poor binders at this receptor subtype (1-methylserotonin or 'Met-I' in the manuscript) and OTV1 with pK_i ≈ 5.5 and 5.1, respectively, according to Figure 5D), especially if compared to the endogenous and reference agonist serotonin (pK_d = 8.9, DOI: 10.1016/0006-2952(95)02122-1). The binding affinity data is only disclaimed in Figure 5, panel D. The pK_i ± SD should be presented upfront, and the results should be analyzed considering this data.

2) With that in mind, the observation on page 5 that "OTV1 has the lowest potency of all compounds in all the pathways monitored" is simply explained by the low affinity of this compound for 5-HT_{2A}. Thus, this molecule does not seem to bring any new insight. I wonder about which criteria were used for the experimental validation of the virtual screening hits, as those are not detailed in the methods (i.e., affinity and/or potency thresholds to consider a ligand a 'hit').

3) I find it particularly intriguing that some of the observed pEC₅₀ values are many orders of magnitude large than the binding affinities. The most dramatic case is Met-I (9.6 for G_q activation, in Table 1) which is 13,000-fold larger than the binding affinity of this compound for 5-HT_{2A}. How is this possible in a (BRET) recruitment/proximity assay that is very little/not affected by downstream signaling amplification? The only explanation, in my point of view, would be a short residence time/fast dissociation rate, which in turn would affect the reception/measurements of biased signaling. It might be useful to measure this molecule's binding kinetics and check how these compare to the time points of data collection, which were not clearly reported. Community guidelines have been made on how to design and report GPCR ligand biases. Please, use them when reporting your data and experimental setup: <https://doi.org/10.1111/bph.15811>

4) The low-affinity Met-I and OTV1 might also be the underlying reason for the complex profiles observed for these compounds in the postmortem brain tissue studies (e.g., G_i inverse agonism), which apparently contradict the results in the cell-based assays (page 7). Moreover, these compounds sharing the core scaffold of serotonin, are likely acting on other 5-HT receptors, the most prominent of them 5-HT_{1A} (a G_{i/o}-coupled receptor), which is probably the second most expressed 5-HT receptor in PFC after 5-HT_{2A} (see <https://doi.org/10.1021/acschemneuro.5b00023> and the expression profiles at <https://pdsp.unc.edu/databases/ShawnCell/home.php>). The 5-HT_{1A} receptor has also been associated with the action of antipsychotic drugs (see <https://www.nature.com/articles/s41398-020-01119-3>). Action upon 5-HT_{1A} likely explains why "selectivity 5-HT_{2A} antagonist does not always reverse the observed effects". Thus, I would suggest that a 5-HT_{1A}-selectivity antagonist would be included in your experiments. Also, as 5-HT_{1A} has also implicated in long-term memory consolidation (see <https://doi.org/10.1016/j.bbr.2020.112932>), this receptor also needs to be taken into consideration in your novel object recognition experiments.

5) The most robust head twitch response observed among the tested compound seems to be for Nitro-I, which is apparently Gq-biased. This is line with recent evidence that HTR is mediated/promoted by Gq activation and/or a multi-pathway signaling efficacy mechanism (see <https://doi.org/10.1021/acscchemneuro.2c00597>, <https://doi.org/10.1101/2023.07.29.551106>, <https://doi.org/10.1016/j.celrep.2023.112203>, and <https://doi.org/10.1016/j.bbrc.2022.01.113>). Met-I and OTV1 do not induce HTR since they bind weakly to 5-HT2A. Based on the pharmacological inhibition of Gq and expression level reduction of Gi proteins for OTV2, the authors claim that HTR is mediated by Gi/o proteins. However, they did not assess this hypothesis for Nitro-I, which also induced a robust HTR, nor the control compound DOI in the same experimental conditions.

6) For all the aforementioned reasons, I consider it particularly difficult to associate the experimental results reported here with specific ligand-receptor interactions or structural determinants via molecular modeling. Position 5 of serotonin' indole ring (and analogs) has been experimentally demonstrated to be a subtype-selectivity determinant for the serotonergic family (see <https://www.nature.com/articles/s41586-021-03376-8> and <https://doi.org/10.1016/j.molcel.2022.05.031>), in particularly due to interaction and/or steric clashes with residue in position 6.55. A simple sequence alignment in GPCRdb (<https://gpcrdb.org/alignment/>) reveals that while 5-HT2A has the bulkier and more hydrophilic Asn343 at position 6.55, 5-HT1A has the smaller and more hydrophobic Ala365. Thus, changing the substituent in the 5-position from nitro/hydroxyl to bulkier/hydrophobic benzodioxin shifts the selectivity towards 5-HT1A and drastically reduces affinity at 5-HT2A (as reported), which might explain the divergent results obtain for the cell-based vs postmortem brain assays. Likewise, the N-methylation (Met-I) likely also shifts the binding profile towards 5-HT1A via clashes with Ser242 (5.46), which is an Alanine in 5-HT1A. This mechanism has been extensively analyzed experimentally for the 5-HT2B receptor (see [doi:10.1038/s41594-018-0116-7](https://doi.org/10.1038/s41594-018-0116-7)). Moreover, roughly 1/3 of the receptor amino acid side chain is missing the template structure (6WHA) and portions of the backbone of ECL2 are also missing. It is not clear how these issues have been addressed in the structure preparation for docking/MD simulation.

Minor:

- a) Figure 2: consider displaying all the 12 dose-response curves (perhaps in a 2x6 panel), also making them larger by better use of space. Please, be careful with the alignment of the graphs (e.g., C vs F)
- b) Table 1: if all dose-response curves are displayed, this table could be moved to SI (double information). However, please, include the data for the reference agonist (serotonin) and I would suggest displaying pEC50 instead of logEC50 to avoid the minus sign.
- c) Figure 2H: the green color is misleading for OTV1, as the ligand does not seem to significantly activate any pathways, rather than being 'Gq-biased' as the color might suggest. This figure is not color-blind safe. Please, consider recoloring to a blue-red scale. You can test the color scales at <https://www.color-blindness.com/coblis-color-blindness-simulator/>

RESPONSE TO REVIEWER COMMENTS

We would like to thank the reviewers for their time and constructive comments, all of which we have thoroughly addressed point by point. Our responses are indicated in blue and new insertions into the manuscript are highlighted in italics. We are confident that the revision has substantially enhanced the overall quality of our manuscript.

Reviewer #1

This is a new submission by Kossatz et al testing the effect of 5-HT_{2A} ligands on G protein coupling and b-arrestin recruitment in cells, as well as functional assays and behavioral phenotypes in mice. The authors proposed that psychedelics induced HTR via Gi proteins, whereas deficits in cognitive performance via Gq. The manuscript is very well written and results are interest since they provide important evidence in support of biased agonist via inhibitory Gi proteins as a key mechanism behind the hallucinogenic properties of psychedelic 5-HT_{2A} agonists.

1- Minor: Fig 2 panel – “dose response” should be changed to “concentration response” since this figure panel shows *in vitro* findings.

We thank the reviewer for pointing out this issue. This has now been corrected.

2- Data *in vitro* are totally unnecessary and confusing since ligands show very different agonist/inverse agonist properties as compared to mouse and human brain samples

We agree with the reviewer that *in vitro* data is not always completely comparable to mouse and human studies. This is indeed a big challenge in multidisciplinary studies. However, we believe it is important to report this data to make researchers aware that extrapolation is not always feasible between different experimental setups. In addition, the observed differences between *in vitro*, *in vivo* and *ex vivo* studies can have important implications for understanding better the etiology of schizophrenia. In other words, understanding better these differences can help guide researchers in future experiments to interrogate the mechanism involved in this disease. Now, we better emphasize these challenges in the discussion section.

“Ultimately, our study highlights the challenges in multidisciplinary research related to the fact that data are often not completely comparable between different experimental setups. Recognizing this divergence in data becomes paramount, as it not only informs researchers about the existing challenge but also serves as a means to elucidate the extent of in vitro, in vivo and human data alignment. In addition, the differences observed between these setups can have significant implications for gaining a better understanding of the disease mechanisms underlying schizophrenia (e.g. diverse expression levels of coupling partners, etc.).”

3- Minor: mouse experiments were conducted in male mice only.

We agree with the reviewer that sex-related experiments should be addressed in future studies, and this has now been better clarified in the methods.

Method: *“Since females have not been tested, this study does not address sex-related differences.”*

4- Demographic data of the postmortem human brain samples are missing in the methods section (sex, age, cause of death etc...)

We thank the reviewer for pointing this out. This information was already included in the Supplementary Material sections. However, we have now moved it to the methods section of the main manuscript to make it more visible to the reader, as follows:

“Samples from 12 different subjects (10 males and 2 females) with ages between 29–90 years were included. The postmortem delay between death and storage of the samples ranged from 4 to 12 h, and the storage time between sampling and experiments ranged from 48 to 10 months.”

5a- Specificity/selectivity of antibodies is usually questionable. How the authors know that Gi1, Gi2, and Gi3 antibodies are specific/selective for this Galpha subunits? Have these antibodies been validated in KO mice?

The reviewer pointed out an important issue when working with antibodies. To test antibody specificity, we performed Western Blot experiments with recombinant antibodies against $G_{\alpha i1}$, $G_{\alpha i2}$, $G_{\alpha i3}$ and $G_{\alpha q/11}$ vs recombinant proteins of G_{α} subunit subtypes and human, rat and mouse brain tissue, and we found that no cross-reactivity exists. We included a new supplementary Figure S8 that reports this finding.

Figure S8. Representative Western-blot images of the selective immunoreactive signal of anti-G α protein subunit subtypes against recombinant G α proteins, and against human, rat and mouse brain membrane homogenates samples. Western-blot images of the selective immunoreactive signal of anti-G α_{i1} (Santa Cruz Biotechnology Cat# sc-56536) (A), anti-G α_{i2} (Santa Cruz Biotechnology Cat# sc-13534) (B), anti-G α_{i3} (Antibodies on-line Cat#ABIN6258933) (C), and anti-G $\alpha_{q/11}$ (Santa Cruz Biotechnology Cat# sc-515689, RRID:AB_2940775) (D) mouse monoclonal antibodies against different recombinant G α proteins. Immunodetection of GNAI1 (His-tagged) (Abeomics Cat# 32-3896; lane 2), GNAI2 (GST-tagged) (Antibodies on-line Cat# ABIN1355337; lane 3), GNAI3 (His-tagged) (Abeomics Cat# 32-3898; lane 4), GNAO (His-tagged) (Antibodies on-line Cat# ABIN5709596; lane 5), GNAQ (His-tagged) (Antibodies on-line Cat# ABIN1355345; lane 6), GNAS (GST-tagged) (Antibodies on-line Cat# ABIN1355349; lane 7), GNAZ (His-tagged) (Cusabio Cat# CSB-EP009601HU; lane 8), and GNA13 (His-tagged) (Cusabio Cat# CSB-EP618885HU; lane 9) recombinant proteins, and human (H), rat (R) and mouse (M) brain cortex membranes is shown. Recombinant proteins were purchased from Abeomics (USA), Cusabio (USA) and Antibodies on-line (Germany). MW: Precision Plus Protein Dual Color Standards molecular weight marker (BioRad). Blue arrow head: 50 kDa. Green arrow head: 37 kDa.

5b- Lack of specificity/selectivity may also explain findings in Fig S4 – for example, it could be that the antiGalpha1 or anti-Galpha3 antibodies are not specific, and therefore knocking down one of these two Galpha proteins leads to reduced immunoreactivity with any of the two antibodies.

To address this question, we incubated both mouse monoclonal anti-G α_{i1} (Santa Cruz Biotechnology Cat# sc-56536) (size 41 kDa) and rabbit polyclonal anti-G α_{i3} (Antibodies on-line Cat#ABIN6258933) (size 45 kDa) in the same nitrocellulose membrane, with anti-mouse Alexa 680 (red in the image) and anti-rabbit IRDye800 (green in the image).

Figure for reviewer. As the reviewer can see, no merging of the signal was observed for recombinant proteins, nor for the membrane homogenates. From this, we conclude that cross-reactivity does not explain the findings in Figure S4A and Figure S4B. In contrast, the lack of specificity of ODN seems to be the most plausible explanation, as has been stated in the Result Section.

6- The authors describe in the Results section that “Met-I and OTV1 elicit inverse agonism at Galphai1...” – This reviewer does not see inverse agonism with Met-I on Galphai1 in Fig 3C.

We thank the reviewer for pointing out this issue. Indeed there was a mistake in the figure arrangement. The plot of OTV1 and OTV2 were erroneously swapped. Now this has been corrected and the reviewer will clearly see the inverse agonism for Met-I on Gai1 compared to OTV2.

7- The description of HTR as a model of positive symptoms in schizophrenia patients is incorrect. HTR is a mouse behavioral proxy of human hallucinogenic potential – this is not modeling “hallucinations” in mice. This part of the text should be corrected.

We thank the reviewer for this important comment. Indeed, the HTR is commonly used as a behavioral proxy in rodents for human hallucinogenic effects and can be used to discriminate hallucinogenic and non-hallucinogenic 5-HT_{2A}R agonists (Gonzalez-Maeso et al. 2007). Therefore, we have corrected the description of HTR in the result section, as follows:

“The HTR serves as a behavioral proxy in rodents for human psychedelic effects, and can be used to discriminate hallucinogenic and non-hallucinogenic 5-HT_{2A}R agonists^{9,20–25}.”

In addition, since HTR is a proxy of the hallucinogenic effects of drugs in humans, and hallucinations are a hallmark of psychosis, we now refer to the induction of HTR by our test compounds as psychosis-related effects throughout the manuscript.

8- Docking in Fig 5 is interesting, but without experimental validation (mutations of the predicted binding pocket, for example) these computer modeling findings are unconvincing.

We thank the reviewer for pointing out this challenging issue.

1. As the reviewer will agree, an unambiguous high-resolution proof of the binding mode for our candidates can only be obtained by X-ray or cryo-EM structures which is beyond the scope of this paper.
2. However, we are highly convinced about the general validity of the proposed binding mode as the main scaffold, i.e. 5-hydroxytryptamine, is confirmed by an

experimentally solved structure (PDB id 7E2Y). The correct placement of this main scaffold allows a confident modeling of the rigid substitutions such as phenoxy or benzodioxin in position 5. In addition, there are numerous mutational studies that probe the binding site of tryptamine, 5-hydroxytryptamine and closely related derivatives further confirming the proposed binding pocket and residues with high contact frequencies observed in our molecule dynamics simulations: V3.33, F6.51 and F6.52.

We agree that this had not been sufficiently reflected in the previous manuscript version. Now, we have improved the result section as follows:

“Structural inspection of the most representative clusters found in simulations reveals that the positioning of the main tryptamine scaffold of all compounds resembles the experimentally solved tryptamine pose of serotonin in the 5-HT_{1A}R (PDB id 7E2Y) with the following interactions of high contact frequencies (Figure 5A-B) (Figure): (i) a salt bridge with D3.32 and (ii) a hydrophobic sandwich formed of V3.33, F6.51 and F6.52. In fact, these positions have been corroborated by numerous mutational studies for tryptamine, 5-hydroxytryptamine and other closely related derivatives (i.e., V3.33³⁶, F6.51 and F6.52^{37–39}. With this central scaffold in place, extension in position 5, as observed in OTV1 (1,4 benzodioxin) and OTV2 (phenoxy), are oriented towards the extracellular loop 2 (ECL2) where they result in increased contact frequencies (Figure 5C).”

9- Back to Fig 2, authors show functional properties but not ligand binding assays. Findings testing affinity (binding displacement curves) of [3H]ketanserin are needed to validate if Nitro-I, Met-I, OTV1 and OTV2 bind to the orthosteric binding pocket of the 5-HT_{2A}

We have noticed that the binding affinity data for our test compounds were not easily accessible in the manuscript, as they had been included at the end of the manuscript in Figure 5D. Now we moved this information to Figure 1. In addition, we have included a sentence at the beginning of the results section stating that our competition binding experiments with [3H]-ketanserin confirm that our test compounds bind to the orthosteric binding site, as follows:

“Competition binding experiments with [³H]-ketanserin confirm that the test compounds bind to the orthosteric binding site of the 5-HT_{2A}R (Figure 1).”

10- Source of the 5-HT_{2A} plasmid (or stable cell line) is not mentioned in the methods section.

This information has been included in the method section, as follows:

“HEK-293 clonal cell lines were a gift from S. Laporte (McGill University, Montreal, Quebec, Canada). The plasmids p63-RhoGEF-RlucII, Rap1Gap-RlucII, β -arrestin1-RlucII,

β -arrestin2-RlucII and rGFP-CAAX have been previously described⁵²⁻⁵⁴ and the human 5-HT_{2A}R was a gift from Domain Therapeutics North America.”

Reviewer #2

In this work, an extensive analysis of the signaling profile of four small molecule agonists of the serotonin 5-HT_{2A} receptor is reported in vitro, in vivo, and ex vivo, followed by in silico rationalization. These molecules – chemically similar to the endogenous agonist serotonin – have been assessed in a cell-based (BRET) assay for their effectiveness in recruiting 12 different effector proteins (from G α q, G α i/o, and β -arrestin families), from which distinct functional bias profiles are reported (two as G α q-biased, one as G α i/o-biased, and one low potency in all pathways). Then, the signaling profiles were investigated in postmortem human brain tissue, where all compounds were found to be 5-HT_{2A}-Gq pathway agonists, but some were found 5-HT_{2A}-Gi inverse agonists while others were agonists in this pathway(s). Then, via experiments with mice (head-twitch response and novel object recognition), the authors conclude that Gi/o signaling mediated by the 5-HT_{2A} receptor promotes pro-psychotic (hallucinogenic) effects while 5-HT_{2A}-mediated Gq signaling enhances memory performance. Finally, these differences in the signaling profile and behavioral responses are linked to specific (and differential) ligand-receptor interactions, in particular to residues of the extracellular loop-2 (ECL2) via computational modeling (molecular dynamics simulations).

I recognize the extensive multidisciplinary effort and the relevance of this work to advance our understanding of the physiopathological roles of the 5-HT_{2A} receptor. In particular, the roles that the multiple signaling pathways elicited by this receptor possess in the behavioral outcomes and its implication, e.g., to the understanding and treatment of schizophrenia and other mental illnesses.

We thank the reviewer for highlighting the importance of our multidisciplinary work and its implication for the treatment of schizophrenia.

However, I have a few concerns about the study that I would like the authors to address before this manuscript can be considered for publication at Nature Communications. I list them below.

1a) Very little attention is given to the fact that two of the four tested molecules are relatively poor binders at this receptor subtype (1-methylserotonin or ‘Met-I’ in the manuscript) and OTV1 with pK_i \approx 5.5 and 5.1, respectively, according to Figure 5D), especially if compared to the endogenous and reference agonist serotonin (pK_d = 8.9, DOI: 10.1016/0006-2952(95)02122-1). The binding affinity data is only disclaimed in Figure 5, panel D. The pK_i \pm SD should be presented upfront, and the results should be analyzed considering this data.

We thank the reviewer for pointing out this issue. We have moved the affinity data to Figure 1 to make them more visible. For result analysis see point 1b.

1b) With that in mind, the observation on page 5 that “OTV1 has the lowest potency of all compounds in all the pathways monitored” is simply explained by the low affinity of this compound for 5-HT2A. Thus, this molecule does not seem to bring any new insight.

Considering the reviewer’s comment, we have realized that the data obtained in our ligand binding experiments primarily reflect binding affinities under conditions in which G proteins are not over-expressed resulting in an apparent low-affinity receptor state for our ligands. In agreement with this notion, in cell-based studies where G proteins are overexpressed (i.e. the receptor shifts to higher affinity states due to G protein coupling), we observed that the ligand-induced G protein activation increases by several orders of magnitude (see point 3 below and corresponding changes in the main text) likely reflecting the increase in ligand binding affinity to the receptor-G protein complex. This can also explain our observation in *ex vivo* studies, where OTV1 is more potent at Gi as an inverse agonist than Nitro-I and OTV2, despite having the lowest affinity in our binding assay.

2) I wonder about which criteria were used for the experimental validation of the virtual screening hits, as those are not detailed in the methods (i.e., affinity and/or potency thresholds to consider a ligand a ‘hit’).

Our study exploited insights from a previous study (Marti Solano *et al.* Mol. Pharm. 2015) where Met-I and Nitro-I showed differential signaling properties on inositol phosphate accumulation (IP) and arachidonic acid release (AA). To study their structure-activity relationship, we selected two additional derivatives with interesting IP and AA profiles (OTV1 and OTV2) via virtual screening and moved on to a comprehensive characterization of their G protein coupling profile and behavioral outcomes. The reference to our previous study (Marti Solano *et al.* Mol. Pharm. 2015) is included in the method section.

3) I find it particularly intriguing that some of the observed pEC50 values are many orders of magnitude larger than the binding affinities. The most dramatic case is Met-I (9.6 for G_{αq} activation, in Table 1) which is 13,000-fold larger than the binding affinity of this compound for 5-HT2A. How is this possible in a (BRET) recruitment/proximity assay that is very little/not affected by downstream signaling amplification? The only explanation, in my point of view, would be a **short residence time/fast dissociation rate**, which in turn would affect the **reception/measurements of biased signaling**.

3a) It might be useful to measure this molecule’s binding kinetics and check how these compare to the time points of data collection, which were not clearly reported.

The reviewer points out a very interesting observation of differences in the affinities determined in radio-ligand binding assays vs the potency in the G_{αq} activity assessed by BRET. Met-I is the most dramatic case with a potency for G_{αq} protein activation being 13,000 fold larger than the ligand binding affinity. The reviewer proposes to carry out ligand binding kinetics. There are two ways of carrying out such ligand-binding kinetics:

(i) We could use a radiolabelled derivative of our test compounds. Unfortunately, this is out of the scope of our study as the generation of radiolabeled ligands is very laborious without guarantee of success.

(ii) An alternative is using a displacement assay of an available radiolabeled ligand. However, this type of experiment is not very reliable as typically one encounters probe dependencies to the used radiolabeled ligand ([10.1016/j.cellsig.2020.109844](https://doi.org/10.1016/j.cellsig.2020.109844)). The reason is that the radiolabeled probe stabilizes a receptor conformational ensemble that is different from the probe-free ensemble, and by doing this it can alter the binding kinetics observed.

However, to address the reviewer's concern and to explore the 13000-fold difference observed between BRET (i.e. pEC50) and binding experiments (i.e. pKi values), first, we titrated the receptor to evaluate if the number of receptors expressed at the plasma membrane, which may have been different in the CHO cells used in the radioligand binding vs the HEK-cells used for the Gq activation assays, could have an impact on the pEC50 of G protein activation. The results show that the quantity of receptor transfected does not affect the observed pEC50 for G protein activation measured at 3 different times indicating that the difference in receptor number is probably not the explanation (new Supplementary Figure S9).

To assess whether the lower affinity detected in the binding assay could result from a probe-dependency (ie: the radio-ligand ketanserin stabilizing the receptor in a low-affinity state for Met-I), we carried out the Met-I-promoted G_{αq} activation assay under similar conditions (ie: in the presence of I or 10 nM ketanserin. As expected, a right shift in the EC50 was observed when increasing the concentration of ketanserin, reflecting the competition between the antagonist and the agonist to bind the 5-HT_{2A}R. The calculated Met-I K_a values ($K_a = EC50 / (1 + [ketanserin] / K_d)$) for G protein activation obtained under these conditions in the BRET experiments for different concentrations of receptor were between 5.7 and 13 nM, values that are very similar to the pEC50 obtained in our original assay in the absence of ketanserin (see Figure below for reviewer). From here we can conclude that the difference between binding affinity and the observed potencies cannot be explained by the stabilization of a low Met-I affinity state by ketanserin.

(B)

	Ketanserin 1 nM			Ketanserin 10 nM		
	EC50	K _a (M)	pK _a	EC50	K _a (M)	pK _a
20ng	1,09E-08	5,71E-09	-8,24	8,25E-08	8,18E-09	-8,09
50ng	1,75E-08	9,17E-09	-8,04	1,30E-07	1,30E-08	-7,89
100ng	1,27E-08	6,68E-09	-8,18	1,08E-07	1,08E-08	-7,97

Figure for reviewer: Competition binding assays of Met-I with ketanserin. (A) Cells were transfected with three different quantities of 5-HT_{2A}R and detection of G_{αq} activation was assessed using the p63RhoGEF-RIucll sensor and the plasma membrane marker (rGFP-CAAX). Following a 10-minute pre-treatment of ketanserin, cells were stimulated with increasing concentrations of Met-I for 10 additional minutes. (B) Table of potency and Ka/pKa for Met-I obtained by competition binding experiments with ketanserin - $K_a = EC50/(1+ [ketanserin] /Kd)$. Data are represented as ligand-promoted BRET (Δ BRET) as mean \pm SEM (n=3).

Another possible explanation for the observed differences in pEC50 (G protein activation) vs pKi (ligand binding) values could be the difference in the amount of G protein expressed in the cells used in the binding assays vs the level in the G_{αq} activation assay. Indeed, in the G_{αq} activation assay, G_{αq} is over-expressed, which is not the case in the binding assay. To test this hypothesis, we carried out additional experiments titrating the heterologous expressed G_{αq} and monitoring the detected pEC50 for G protein activation. We performed these experiments in both HEK-293 and CHO cells. As can be seen in Supplementary Figure S10, we found that increasing the levels of G_{αq} results in a left shift (higher potency) of the Met-I-induced G_{αq} activation curve. Moreover, Met-I potency was found to be higher in HEK vs CHO cells, consistent with the notion that the over-expressed G protein most likely explains the high potency observed in this assay. Most importantly, we further find that the pEC50 for G protein activation using only endogenous G_{αq} (pEC50 6.87) in CHO cells is closer to the ligand binding affinity of Met-I (pKi 5.45) found in the radio-ligand binding assay. This finding demonstrates that the level of G protein is a critical parameter underlying the differences observed between pEC50 of G protein activation and pKi of ligand binding.

We have now added these findings to the results section as follows:

“An interesting observation is that some of the observed pEC50 values are many orders of magnitude larger than the corresponding binding affinities (Figure 1B). An example is Met-I with a pEC50 9.6 for G_{αq} activation (Table 1) versus a pKi 5.5 for its binding affinity to the 5-HT_{2A}R (Figure 1). This difference most likely results from the diverse experimental setups including the expression level of receptors or G proteins¹⁵. Interestingly, additional BRET experiments show that receptor expression levels do not affect the observed pEC50 values (Figure S9). In contrast, we find that increasing levels of G_{αq} significantly increment the apparent potency of Met-I (Figure S10). This finding demonstrates that the expression level of G protein is a critical parameter that largely contributes to the differences observed between pEC50 of G protein activation and pKi of ligand binding. It further underscores the difficulties associated with comparing data points across distinct experimental setups and could also explain the differences observed between cell-based and ex vivo experiments.”

3b) Community guidelines have been made on how to design and report GPCR ligand biases. Please, use them when reporting your data and experimental setup: <https://doi.org/10.1111/bph.15811>

We have carefully revised our manuscript and made sure that we report our data according to the community guidelines.

4a) The low-affinity Met-I and OTV1 might also be the underlying reason for the complex profiles observed for these compounds in the postmortem brain tissue studies (e.g., Gi inverse agonism), which apparently contradict the results in the cell-based assays (page 7).

As discussed in point 3a, the presence of higher expression levels of G protein dramatically increases by different magnitudes the compound-induced G protein activation. Therefore, we believe that differential G protein expression levels in the cell-based and postmortem brain approaches could explain the contradictory results. This is now also stated in the result section (see point 3a).

“ ... It further underscores the difficulties associated with comparing data points across distinct experimental setups and could also explain the differences observed between cell-based and ex vivo experiments.”

4b) Moreover, these compounds sharing the core scaffold of serotonin, are likely acting on other 5-HT receptors, the most prominent of them 5-HT_{1A} (a Gi/o-coupled receptor), which is probably the second most expressed 5-HT receptor in PFC after 5-HT_{2A} (see <https://doi.org/10.1021/acchemneuro.5b00023> and the expression profiles at <https://pdsp.unc.edu/databases/ShawnCell/home.php>) the 5-HT_{1A} receptor has also been associated with the action of antipsychotic drugs (see <https://www.nature.com/articles/s41398-020-01119-3>). Action upon 5-HT_{1A} likely explains why “selectivity 5-HT_{2A} antagonist does not always reverse the observed effects”. Thus, I would suggest that a 5-HT_{1A}-selectivity antagonist would be included in your experiments. Also, as 5-HT_{1A} has also implicated in long-term memory consolidation (see <https://doi.org/10.1016/j.bbr.2020.112932>), this receptor also needs to be taken into consideration in your novel object recognition experiments.

We would like to thank this reviewer for pointing out this issue. As this reviewer describes, 5-HT_{1A}R are highly expressed in the PFC and have been implicated in antipsychotic effects, and memory processes. However, the effects induced by our studied compounds for pathways linked to cognitive deficits (G_{αq}) and HTR (G_{αi1}) in *ex vivo* experiments are completely reversed by the 5-HT_{2A}R-selective antagonist MDL 11,939. Similarly, these effects are absent in those assays carried out with membrane homogenates from 5-HT_{2A}R KO mice brain cortex. Both results demonstrate the selective role of 5-HT_{2A}R in these effects. Likewise, the HTR and the novel object recognition deficits induced by these compounds are abrogated in 5-HT_{2A}R knockout mice, supporting the crucial involvement of these receptors. In addition, previous studies carried out with psilocybin (a tryptamine derivative) have also discarded the involvement of 5-HT_{1A}R in HTR (10.1016/j.biopha.2022.113612). As the reviewer would agree, based on this evidence, studying the implications of 5-HT_{1A}R in those pathways as a main contributor is not necessary.

5a) The most robust head twitch response observed among the tested compound seems to be for Nitro-I, which is apparently Gq-biased. This is line with recent evidence that HTR is mediated/promoted by Gq activation and/or a multi-pathway signaling efficacy mechanism (see <https://doi.org/10.1021/acchemneuro.2c00597>, <https://doi.org/10.1101/2023.07.29.551106>, <https://doi.org/10.1016/j.celrep.2023.112203>, and <https://doi.org/10.1016/j.bbrc.2022.01.113>). Based

on the pharmacological inhibition of Gq and expression level reduction of Gi proteins for OTV2, the authors claim that HTR is mediated by Gi/o proteins. However, they did not assess this hypothesis for Nitro-I, which also induced a robust HTR, nor the control compound DOI in the same experimental conditions.

This reviewer is correct, we did not assess whether Nitro-I or DOI-induced HTR was mediated by $G_{\alpha i/o}$ proteins in our study. Instead, we selected one compound with the most interesting behavioral profile, i.e., OTV2, which induces both HTR and cognitive deficits. This was done to reduce the number of animals used in the study, complying with current reduction policies for animal experimentation.

However, regarding DOI, the involvement of $G_{\alpha i/o}$ in HTR is further supported by our unpublished data (see Figure below) showing that administration of the selective $G_{\alpha i/o}$ inhibitor, PTX (0.4 μ g, ICV) significantly reduced (\pm)-DOI-induced HTR (see Figure A and B below), while it had no effect over hyperthermia. On the other hand, pretreatment with the selective $G_{\alpha q}$ inhibitor, YM-254890 (16 μ M, ICV) had no significant effect over (\pm)-DOI-induced HTR, while it prevented the hyperthermia response (see Figure C-D below). Thus, in our hands, DOI-induced HTR is $G_{\alpha i/o}$ - and not $G_{\alpha q}$ -mediated.

Nevertheless, we cannot completely exclude other mechanisms than $G_{\alpha i/o}$ activation in mediating HTR. We acknowledge this in the main manuscript as follows:

“However, we cannot completely exclude mechanisms different from $G_{\alpha i/o}$ activation in mediating HTR. Previous studies have described the involvement of other coupling partners including $G_{\alpha q}$ ^{25,27–30}, although there are also studies showing that $G_{\alpha q}$ KO mice inhibit only partially HTR²⁷, suggesting additional contributing mechanisms in HTR. Moreover, studies have reported the involvement of $G_{\alpha s}$ proteins³⁰, $G_{\beta\gamma}$ subunits³¹, and β -arrestins^{11,12} in HTR.”

5b) This referee also points out that Met-I and OTV1 may not induce HTR because they bind weakly to 5-HT2AR.

We would like to refer the reviewer to point 3a, where we state that despite Met-I and OTV1 having low affinity in ligand binding assays (basal expression levels of G proteins), they show potent G protein activation in the presence of G protein in cell-based assays. In addition, we carried out *in vivo* dose-response experiments showing that the administration of progressively higher doses of these compounds does not increase the presence of HTR. Moreover, assays in postmortem brain tissue show how these “low-affinity ligands” are able to trigger significant inhibitory/stimulatory effects over different G_α subunits. Therefore, we do not believe that low binding affinities are responsible for the lack of HTR.

6a) For all the aforementioned reasons, I consider it particularly difficult to associate the experimental results reported here with specific ligand-receptor interactions or structural determinants via molecular modeling.

Position 5 of serotonin' indole ring (and analogs) has been experimentally demonstrated to be a subtype-selectivity determinant for the serotonergic family (see <https://www.nature.com/articles/s41586-021-03376-8> and <https://doi.org/10.1016/j.molcel.2022.05.031>), in particular due to interaction and/or steric clashes with residue in position 6.55. A simple sequence alignment in GPCRdb (<https://gpcrdb.org/alignment/>) reveals that while 5-HT_{2A} has the bulkier and more hydrophilic Asn343 at position 6.55, 5-HT_{1A} has the smaller and more hydrophobic Ala365. Thus, changing the substituent in the 5-position from nitro/hydroxyl to bulkier/hydrophobic benzodioxin shifts the selectivity towards 5-HT_{1A}R and drastically reduces affinity at 5-HT_{2A} (as reported), which might explain the divergent results obtain for the cell-based vs postmortem brain assays. Likewise, the N-methylation (Met-I) likely also shifts the binding profile towards 5-HT_{1A} via clashes with Ser242 (5.46), which is an Alanine in 5-HT_{1A}. This mechanism has been extensively analyzed experimentally for the 5-HT_{2B} receptor (see doi:10.1038/s41594-018-0116-7).

We very much appreciate the considerations of the reviewer, and we are aware that 5-HT_{1A}R could be an important contributor to the effects of our compounds. In fact, we have binding data that support that our compounds can bind to the 5-HT_{1A}R. However, as previously discussed in point 4, studies with psilocybin (tryptamine-related psychedelic compound) have demonstrated that even though this compound shows good affinity for 5-HT_{1A}R, its psychedelic-like effects are not mediated through this receptor (10.1016/j.biopha.2022.113612).

Moreover, we show that cognitive deficits and HTR induced by our tryptamine derivatives are completely mediated by the 5-HT_{2A}R (KO experiments and 5HT_{2A}R-specific inhibition by MDL; see a more complete discussion in point 4b). This strongly suggests that ligand-specific interactions with the 5-HT_{2A}R are responsible for the observed effects. Therefore, we believe that modeling 5HT_{1A}R complexes would not add further value to the obtained results.

6b) Moreover, roughly 1/3 of the receptor amino acid side chain is missing the template structure (6WHA) and portions of the backbone of ECL2 are also missing. It is not clear how these issues have been addressed in the structure preparation for docking/MD simulation.

We thank the reviewer for pointing out this missing information. We have now included it in the methods section.

Minor:

a) Figure 2: consider displaying all the 12 dose-response curves (perhaps in a 2x6 panel), also making them larger by better use of space. Please, be careful with the alignment of the graphs (e.g., C vs F)

We have considered the reviewer's suggestions. To obtain an acceptable graph size, we have increased the plot size and decided to plot only 6 dose-response curves. In addition, we have fixed the alignment for C and F.

b) Table 1: if all dose-response curves are displayed, this table could be moved to SI (double information). However, please, include the data for the reference agonist (serotonin) and I would suggest displaying pEC50 instead of logEC50 to avoid the minus sign.

We have included the serotonin data and changed logEC50 to pEC50.

c) Figure 2H: the green color is misleading for OTV1, as the ligand does not seem to significantly activate any pathways, rather than being 'Gq-biased' as the color might suggest. This figure is not color-blind safe. Please, consider recoloring to a blue-red scale. You can test the color scales at <https://www.color-blindness.com/coblis-color-blindness-simulator/>

We thank the reviewer for pointing out the color blind issue. We have now adapted the color scale to red-white-blue. Furthermore, we have substituted the green color of OTV1 by gray to better emphasize that the G_{aq} bias could not be computed.

Reviewer #1 (Remarks to the Author):

The authors have addressed all previous comments satisfactorily. My only minor concern is that [³H]ketanserin binding displacement findings are not presented in Fig 1 (as mentioned in the results section and in the response to the reviewers).

Reviewer #2 (Remarks to the Author):

The authors have addressed my comments and concerns satisfactorily. The manuscript can be recommended for publication.

Minor:

- Table 1: The "minus sign" in the pEC₅₀ values of Gai1 and Gaz for OTV1 might be incorrect.
- Page 5 line 24: "ligand physiology" is repeated.

REVIEWERS' COMMENTS

Reviewer #1 (Remarks to the Author):

The authors have addressed all previous comments satisfactorily.

My only minor concern is that [³H]ketanserin binding displacement findings are not presented in Fig 1 (as mentioned in the results section and in the response to the reviewers).

This information is presented in Fig. 1. We have modified the figure legend to clarify this point, as follows: "Structural derivatives of the endogenous 5-HT_{2A}R agonist serotonin (5-HT). Ligand binding affinities (pKi) are indicated for Nitro-I, Met-I, OTV1 and OTV2 obtained in [³H]ketanserin competition binding experiments in CHO cells (n=3). The data represent the mean ± SD (see methods section)".

Reviewer #2 (Remarks to the Author):

The authors have addressed my comments and concerns satisfactorily. The manuscript can be recommended for publication.

Minor:

- Table 1: The "minus sign" in the pEC50 values of Gai1 and Gaz for OTV1 might be incorrect.

This mistake has been corrected.

- Page 5 line 24: "ligand physiology" is repeated.

This mistake has been corrected.